# LENGTH DESENSITIZATION IN DIRECT PREFERENCE OPTIMIZATION

## ABSTRACT

Direct Preference Optimization (DPO) is widely utilized in the Reinforcement Learning from Human Feedback (RLHF) phase to align Large Language Models (LLMs) with human preferences, thereby enhancing both their harmlessness and efficacy. However, it has been observed that DPO tends to over-optimize for verbosity, which can detrimentally affect both performance and user experience. In this paper, we conduct an in-depth theoretical analysis of DPO's optimization objective and reveal a strong correlation between its implicit reward and data length. This correlation misguides the optimization direction, resulting in *length sensitivity* during the DPO training and leading to verbosity. To address this issue, we propose a length-desensitization improvement method for DPO, termed LD-DPO. The proposed method aims to desensitize DPO to data length by decoupling explicit length preference, which is relatively insignificant, from the other implicit preferences, thereby enabling more effective learning of the intrinsic preferences. We utilized two settings (Base and Instruct) of Llama2-13B, Llama3-8B, and Qwen2-7B for experimental validation on various benchmarks including MT-Bench and AlpacaEval 2. The experimental results indicate that LD-DPO consistently outperforms DPO and other baseline methods, achieving more concise responses with a 10-40% reduction in length compared to DPO. We conducted in-depth experimental analyses to demonstrate that LD-DPO can indeed achieve length desensitization and align the model more closely with human-like preferences.

*"Brevity is the Soul of Wit."*

*—William Shakespeare*

## 1 INTRODUCTION

Large Language Models (LLMs) have revolutionized Natural Language Processing (NLP) by empowering machines to generate human-like text, comprehend intricate context, and execute a wide range of linguistic tasks with unprecedented accuracy(Ouyang et al., 2022; Chang et al., 2024; Liu et al., 2023). Aligning LLMs with human values and preferences through learning from human feedback is crucial to ensuring these models are helpful, honest, and harmless. Among the various methods to achieve effective alignment (Dai et al., 2024; Yuan et al., 2024a), Direct Preference Optimization (DPO) has emerged as a promising technique (Rafailov et al., 2024), giving rise to numerous derivative algorithms (Hong et al., 2024; Chen et al., 2024b; Ethayarajh et al., 2024). DPO eliminates the reliance on online Reward Models (RMs) by reparameterizing the reward function in Reinforcement Learning from Human Feedback (RLHF), thereby implementing a simple and stable offline preference learning paradigm. Among the dimensions of human language preferences, detailedness is one of the most straightforward categories that current alignment algorithms can effortlessly capture, as longer texts tend to be richer in content. However, it has

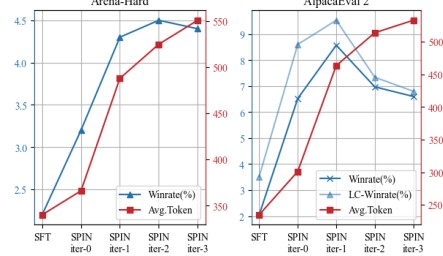

Figure 1: Performance of iterative DPO model(Chen et al., 2024d) on Arena-Hard and Alpacaeval 2.

been demonstrated that DPO is susceptible to an over-optimization issue in this particular preference dimension (Xu et al., 2024). As illustrated in Fig.1, the model post-DPO tends to generate longer responses. However, excessively long responses can result in performance degradation, particularly impacting its instruction-following and reasoning capabilities (Ding et al., 2023; Yuan et al., 2024b).

The phenomenon of verbose response caused by DPO is often attributed to the presence of length bias in the training data (Park et al., 2024; Singhal et al., 2023). This bias arises from an inherent length preference in offline RMs (Wang et al., 2024; Chen et al., 2024c), which results in most preferred responses (*chosen*) being significantly longer than the dispreferred ones (*rejected*). Based on this assumption, Yuan et al. (2024b) proposed LIFT-DPO to mitigate the length bias in the training data through a prompt enhancement strategy. Recently, more researchers have questioned the efficacy of the DPO algorithm itself. Park et al. (2024) introduce a regularization term in the optimization objective to adjust the weight of the gradient according to the length difference between the preference pairs. Meng et al. (2024) propose a reference-model-free method SimPO, which used the likelihood averaged by length to eliminate the effect of data length. Lu et al. (2024) introduce a down-sampling approach on KL divergence to eliminate the length reliance of DPO. Though Lu et al. (2024) have conducted a statistical analysis of the implicit rewards during the DPO process and found that the rewards might be overestimated or underestimated due to length, the theoretical explanation of why DPO encounters this issue remains inadequately explored. Meanwhile, experimental results demonstrate that these methods either fail to achieve significant length control or compromise the performance to some extent.

In this paper, we attribute the verbosity problem to the *length sensitivity* of DPO. Specifically, the partial derivatives of the optimization objective of DPO with respect to the *chosen* and *rejected* responses are inversely proportional to their respective likelihood (Feng et al., 2024a). Since the likelihood, which is calculated as the product of the conditional probabilities of each token, decreases rapidly with increasing sequence length, longer *chosen* or *rejected* responses are disproportionately favored in the optimization process. Moreover, the length disparity between the *chosen* and *rejected* responses will substantially skew the optimization objective, ultimately biasing the direction of optimization. Since decreasing the likelihood of any *rejected* may not affect response length, the primary cause of the verbosity problem is the model's tendency to increase the likelihood of longer *chosen* responses while ignoring shorter ones.

To address this issue, we propose an offline optimization algorithm for **L**ength **D**esensitization of **DPO**, termed **LD-DPO**. In this approach, we decompose the likelihood of the longer response in a preference pair into the product of the likelihood of the public-length portion and the likelihood of the excessive portion. The excessive portion is further broken down into verbosity preference (due to excess length) and other preferences. LD-DPO aims to mitigate the verbosity preference caused by excessively long responses, thereby smoothing the relationship between the likelihood and response length. This adjustment reduces the influence of length on the optimization direction in DPO, effectively achieving length desensitization.

We employ two settings (Base and Instruct) of Llama2-13B (Touvron et al., 2023), Llama3-8B (AI@Meta, 2024), and Qwen2-7B (Yang et al., 2024) for experimental validation on various benchmarks including MT-Bench (Zheng et al., 2024) and AlpacaEval 2 (Dubois et al., 2024). The experimental results indicate that LD-DPO consistently outperforms DPO and other baseline methods, achieving more concise responses with a 10-40% reduction in length compared to DPO. Moreover, experiment on reasoning-specific benchmarks shows that LD-DPO significantly improves models' reasoning performance. An interesting phenomenon is also observed: the *length sensitivity* during DPO training exhibits a negative correlation with the underlying model capability. we then define $\gamma$ as the *length sensitivity coefficient* and conduct a detailed analysis of the DPO *length sensitivity* across models of varying capabilities, we believe $\gamma$ is instructive for the entire preference optimization process. Our contributions are summarized as follows:

- To the best of our knowledge, we are the first to define the *length sensitivity* of DPO and provide theoretical validation for this phenomenon.

- We propose LD-DPO, a length-desensitization preference optimization algorithm that mitigates *length sensitivity* by decoupling length preference from the reward.

- We experimentally verify that LD-DPO enables the model to achieve superior results with more concise responses, reducing response length by 10-40% compared to DPO.

## 2 PRELIMINARIES

In this section, we will outline the standard pipeline of Reinforcement Learning From Human Feedback (RLHF) (Bai et al., 2022; Ziegler et al., 2019) and the Direct Preference Optimization (DPO) algorithm (Rafailov et al., 2024), which is essential for the analysis of the *length sensitivity* of DPO and the motivation of our method.

### 2.1 REINFORCEMENT LEARNING FROM HUMAN FEEDBACK

The standard pipeline of RLHF aligns LLMs with human preferences in three stages:

**Supervised Fine-tuning (SFT) stage:** In this stage, labeled data is used to fine-tune the pre-trained model so that it acquires a basic ability to follow commands and carry on a fluent conversation, to obtain model $\pi^{SFT}(y|x)$.

**Reward Model (RM) Training stage:** In the second stage, $\pi^{SFT}(y|x)$ is utilized by prompts $x$ to generate pairs of responses $(y_1, y_2) \sim \pi^{SFT}(y|x)$, which are then labeled by human annotators as a preferred answer $y_w$ and a dispreferred answer $y_l$, denoted as $y_w > y_l$. To predict these preferences, previous works employ the Bradley-Terry (BT) RM (Bradley & Terry, 1952), which essentially constructs a pairwise contrast:

$$\mathcal{L}_{RM} = -\log \frac{exp(r_\phi(x, y_w))}{exp(r_\phi(x, y_w)) + exp(r_\phi(x, y_l))}. \tag{1}$$

**Reinforcement Learning (RL) stage:** In the Final Stage, the reward function is used to provide feedback to the language model. The optimization objective is formulated as:

$$\max_{\pi_\theta} \mathbb{E}_{x \sim \mathcal{D}, y \sim \pi_\theta(y|x)}[r_\phi(x, y)] - \beta \mathbb{D}_{KL}[\pi_\theta(y|x) \| \pi_{ref}(y|x)], \tag{2}$$

where $\beta$ is a parameter controlling the deviation from the reference model $\pi_{ref}$, namely the initial SFT model $\pi^{SFT}(y|x)$, and in practice, the language model $\pi_\theta$ is also initialized to $\pi^{SFT}(y|x)$. This objective is optimized using a general purpose RL algorithm, such as PPO (Wu et al., 2023).

### 2.2 DIRECT PREFERENCE OPTIMIZATION

Direct Preference Optimization (DPO) is one of the most popular offline preference optimization methods, which starts with the same objective as Eq.2, reparameterizes the reward function $r$ using a closed-form expression with the optimal policy:

$$r(x, y) = \beta \log \frac{\pi_\theta(y|x)}{\pi_{ref}(y|x)} + \beta \log Z(x), \tag{3}$$

where $Z(x) = \sum_y \pi_{ref}(y|x) exp(\frac{1}{\beta} r(x, y))$ is the partition function, which is only relevant for $\pi_{ref}$ and $\pi_\theta$, no additional training of the RM is required. By incorporating Eq.3 into the BT ranking objective, $p(y_w > y_l|x) = \sigma(r(x, y_w) - r(x, y_l))$, therefore, the optimization objective becomes:

$$\mathcal{L}_{DPO}(\pi_\theta; \pi_{ref}) = -\mathbb{E}_{(x, y_w, y_l) \sim \mathcal{D}}[\log \sigma(\beta \log \frac{\pi_\theta(y_w|x)}{\pi_{ref}(y_w|x)} - \beta \log \frac{\pi_\theta(y_l|x)}{\pi_{ref}(y_l|x)})]. \tag{4}$$

DPO replaces the reward model (RM) with an implicit reward, offering enhanced stability and ease of training compared to traditional reinforcement learning methods such as PPO. Several related works have validated the effectiveness of this paradigm.

## 3 METHODOLOGY

In this section, we first conduct a theoretical analysis of the optimization object of DPO and verify that differences in data length significantly affect the optimization direction during the training process, demonstrating that DPO is length-sensitive. We then derive our LD-DPO algorithm, which addresses the *length sensitivity* problem by reparameterizing the likelihood, thereby preventing the generation of verbose responses and aligning the model more closely with human-like preferences.

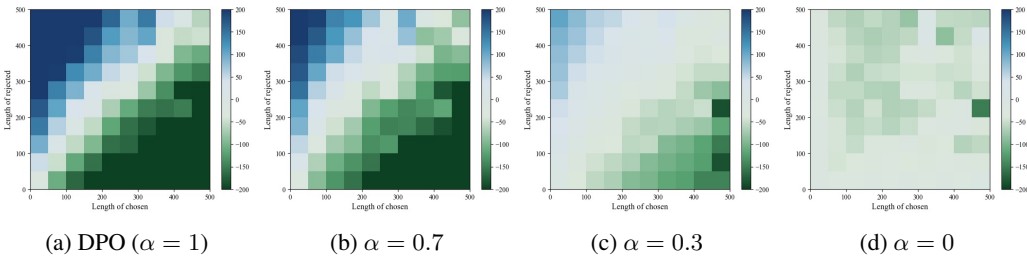

(a) DPO ($\alpha = 1$)  (b) $\alpha = 0.7$  (c) $\alpha = 0.3$  (d) $\alpha = 0$

Figure 2: Comparison of the relationship between the length of preference data pairs and $\pi_\theta(y_l|x)/\pi_\theta(y_w|x)$ under both DPO and LD-DPO. Measured on Llama3-8B-Instruct with Ultra-Feedback dataset (Cui et al., 2023), and the heatmap values represent $\log \pi_\theta(y_l|x) - \log \pi_\theta(y_w|x)$.

## 3.1 LENGTH SENSITIVITY OF DPO

According to the optimization objective of DPO in Eq.4, we know that the purpose of DPO is to make the likelihood of human-preferred response $y_w$ given $x$ greater than that of human-dispreferred response $y_l$, denoted as $\pi_\theta(y_w|x) > \pi_\theta(y_l|x)$. Additionally, $\pi_\theta$ serves as the actor model, while $\pi_{ref}$ is introduced to prevent the model from deviating from the reference model. Both models are tpyically products of the Supervised Fine-Tuning (SFT) phase.

Following Feng et al. (2024a), we provide the theoretical derivation of the optimization objective for DPO. Firstly, we denote $\mathcal{X}_1 = \pi_\theta(y_w|x)$, $\mathcal{X}_2 = \pi_\theta(y_l|x)$, $\mathcal{K}_1 = \pi_{ref}(y_w|x)$, $\mathcal{K}_2 = \pi_{ref}(y_l|x)$ and assuming that the expectation sign is removed in the case of identically distributed training data, the loss function of DPO can be written as:

$$\mathcal{L}_{DPO}(\mathcal{X}_1; \mathcal{X}_2) = -\log\left(\frac{(\mathcal{K}_2\mathcal{X}_1)^\beta}{(\mathcal{K}_2\mathcal{X}_1)^\beta + (\mathcal{K}_1\mathcal{X}_2)^\beta}\right). \tag{5}$$

We calculate the partial derivatives of $\mathcal{L}_{DPO}$ with respect to $\mathcal{X}_1$, and $\mathcal{X}_2$:

$$\begin{cases} \dfrac{\partial \mathcal{L}_{DPO}(\mathcal{X}_1; \mathcal{X}_2)}{\partial \mathcal{X}_1} = -\dfrac{\beta(\mathcal{K}_1\mathcal{X}_2)^\beta}{\mathcal{X}_1((\mathcal{K}_2\mathcal{X}_1)^\beta + (\mathcal{K}_1\mathcal{X}_2)^\beta)}, \\ \dfrac{\partial \mathcal{L}_{DPO}(\mathcal{X}_1; \mathcal{X}_2)}{\partial \mathcal{X}_2} = \dfrac{\beta\mathcal{K}_1^\beta \mathcal{X}_2^{\beta-1}}{(\mathcal{K}_2\mathcal{X}_1)^\beta + (\mathcal{K}_1\mathcal{X}_2)^\beta}, \end{cases} \tag{6}$$

then leading to the following result:

$$\left| \frac{\partial \mathcal{L}_{DPO}(\mathcal{X}_1; \mathcal{X}_2)}{\partial \mathcal{X}_1} \middle/ \frac{\partial \mathcal{L}_{DPO}(\mathcal{X}_1; \mathcal{X}_2)}{\partial \mathcal{X}_2} \right| = \frac{\mathcal{X}_2}{\mathcal{X}_1} = \frac{\pi_\theta(y_l|x)}{\pi_\theta(y_w|x)}. \tag{7}$$

Therefore, the partial derivatives of the optimization objective with respect to $\pi_\theta(y_w|x)$ and $\pi_\theta(y_l|x)$ are inversely proportional to their respective values. Furthermore, the derivation process of Eq.7 and a detailed analysis of the absolute magnitude of the gradient is provided in Appendix A.1. When $\pi_\theta(y_w|x)$ is less than $\pi_\theta(y_l|x)$, DPO tends to increase the likelihood of generating human-preferred response $y_w$. Conversely, DPO tends to avoid generating human-dispreferred response $y_l$. Based on this conclusion, we will analyze the *length sensitivity* of DPO as follows.

In DPO process, the likelihood $\pi_\theta(y|x)$ for sequence-level output is obtained by cumulatively multiplying the conditional probability of each token $p(y_t|x, y_{<t})$ as shown in Eq.8.

$$\pi_\theta(y|x) = \prod_{i=1}^{len(y)} p(y_i|x, y_{<i}). \tag{8}$$

As the conditional probability of the current token from the policy $p(y_t|x, y_{<t})$ lies within the range $[0, 1]$, it follows that as the sentence $y$ consists of more tokens, $\pi_\theta(y|x)$ will obviously be smaller. According to Eq.7, we know that if $len(y_w) > len(y_l)$, then it is highly likely that $\pi_\theta(y_w|x) < \pi_\theta(y_l|x)$, so the language model tends to generate the longer response $y_w$ after DPO; Conversely, if $len(y_w) < len(y_l)$, DPO prevents the output of the longer answer $y_l$, but this does not imply that the shorter answer $y_w$ will be preferred, then verbosity arises.

As shown in Fig.2a and based on the above analysis, DPO is more sensitive to data pairs with large differences in length. Therefore, it tends to guide the model to prioritize length preferences in the data, ignoring other human-like preferences that are more important.

## 3.2 DERIVATION OF LD-DPO

Based on the analysis conducted in the preceding section, it is evident that the *length sensitivity* of DPO primarily originates from the substantial influence text length exerts on the likelihood $\pi_\theta(y|x)$. This influence consequently biases the optimization process towards favoring data with a length advantage. To address this issue, Length-Desensitization DPO (LD-DPO) is employed to diminish the impact of length on the likelihood. This adjustment allows the optimization process to focus more on the substantive content of the text, thereby better aligning with human preference.

For a pair of preference data $(y_w, y_l)$ with lengths $(l_w, l_l)$, we denote $l_p = min(l_w, l_l)$ as the public length. Then the likelihood of response in DPO can be rewritten as:

$$\pi_\theta(y|x) = \prod_{i=1}^{l_p} p(y_i|x, y_{<i}) \prod_{i=l_p+1}^{l} p(y_i|x, y_{<i}), \qquad (9)$$

where $l$ is the length of $y$. The second term contains extensive length information, which directly decreases the reward and further biases the optimization objective. This bias contributes to the overall *length sensitivity* of DPO.

In LD-DPO, our objective is to attenuate the sensitivity of DPO by eliminating the verbosity preferences induced by the excessively long portions, while concurrently maintaining the other preferences, which include a certain degree of length preference. Initially, as demonstrated in Eq.10, we disassociate the verbosity preferences from the likelihood of over-long portion (second term in Eq.9) by introducing a hyperparameter $\alpha \in [0, 1]$.

$$\prod_{i=l_p+1}^{l} \underbrace{p^\alpha(y_i|x, y_{<i})}_{\text{other preferences}} \underbrace{p^{1-\alpha}(y_i|x, y_{<i})}_{\text{verbosity preference}}. \qquad (10)$$

We then diminish the *length sensitivity* of DPO by removing verbosity preference from $\pi_\theta(y|x)$, obtaining the modified likelihood $\hat{\pi}_\theta(y|x)$ in LD-DPO:

$$\hat{\pi}_\theta(y|x) = \prod_{i=1}^{l_p} p(y_i|x, y_{<i}) \prod_{i=l_p+1}^{l} p^\alpha(y_i|x, y_{<i}). \qquad (11)$$

When $\alpha = 1$, $\hat{\pi}_\theta(y|x) = \pi_\theta(y|x)$, which is consistent with vanilla DPO. Conversely, when $\alpha = 0$, the likelihood of over-length part is equal to 1, meaning that only the public-length part will be considered. Ultimately, we reformulate $\hat{\pi}_\theta(y_k|x)$ in Eq.12 to present it in a more elegant form, with the detailed derivation provided in Appendix A.2.

$$\hat{\pi}_\theta(y|x) = \prod_{i=1}^{l} p^\alpha(y_i|x, y_{<i}) \prod_{i=1}^{l_p} p^{1-\alpha}(y_i|x, y_{<i}). \qquad (12)$$

It is observable that $\hat{\pi}_\theta(y|x)$ is constituted by the complete sequence and the public-length component of the preference data pair. The proportion between these two components can be modulated by adjusting the hyperparameter $\alpha$. In scenarios where the *length sensitivity* during the DPO training process is relatively pronounced, a smaller $\alpha$ should be opted for in order to decouple the verbosity preference. Conversely, a larger $\alpha$ should be selected to avert the loss of genuine human preferences.

As shown in Fig.2, compared to DPO (Fig.2a), LD-DPO (Fig.2b, 2c, 2d)) with any $\alpha \in [0, 1)$ can smooth $\pi_\theta(y_l|x)/\pi_\theta(y_w|x)$, and this effect is markedly amplified as $\alpha$ diminishes. Based on the prior analysis and Eq.7, it is evident that the optimization direction of LD-DPO is less affected by the length disparity within the preferred data pairs. This indicates that, relative to DPO, LD-DPO has achieved a measure of length desensitization.

## 4 Experimental Setup

We follow the experimental setup of SimPO (Meng et al., 2024) to objectively demonstrate the validity of our method.

**Models and training settings.** We perform preference optimization using three families of models: Llama2-13B (Touvron et al., 2023), Llama3-8B (AI@Meta, 2024) and Qwen2-7B (Yang et al., 2024) under two setups: Base and Instruct/Chat.

For the **Base** setup, we train a base language model on the UltraChat-200k dataset (Ding et al., 2023) to obtain an SFT model, which possesses a basic capability for conversation. For the **Instruct** setup, we select their corresponding instruct models (i.e., Llama2-13B-Chat, Llama3-8B-Instruct, and Qwen2-7B-Instruct) as initial models. These models are more powerful and robust compared to the base models. Both setups ensure a high level of transparency as the models and training data are open source.

In the preference optimization phase, we utilize UltraFeedback(Cui et al., 2023) as the human preference dataset. This dataset consists of 60,000 high-quality data pairs $(x, y_w, y_l)$ designed to align with human conversational preferences and emphasize helpfulness.

**Evaluation benchmarks.** We primarily evaluate our models using three of the most popular open-ended evaluation benchmarks: MT-Bench (Zheng et al., 2024), AlpacaEval 2 (Dubois et al., 2024) and Arena-Hard (Li et al., 2024). These benchmarks assess the model's versatile session capabilities across a wide range of queries and have been widely adopted by the community.

MT-Bench comprises 80 questions spanning 8 categories, whereas AlpacaEval 2 encompasses 805 questions derived from 5 datasets. We present the results in accordance with the evaluation protocol designated for each benchmark. For MT-Bench, we present the average score and provide a detailed breakdown of the scores for each capability item in Appendix B.3. In the case of AlpacaEval 2, we report the length-controlled (LC) win rate against GPT-4-preview-1106, a metric specifically engineered to be resistant to model verbosity. For space reasons, we present the analysis of Arena-Hard in Appendix B.1 All our evaluations are executed utilizing GPT4-turbo-0409 as the adjudicating model. Furthermore, we calculate the average response length on each benchmark to compare the effects of different methods on response length.

**Baselines.** We compare LD-DPO with five other offline preference optimization techniques. Among these, DPO serves as our most crucial comparison. R-DPO revises DPO by incorporating a length regularity term to control the response length. SamPO avoides reward overestimation or underestimation due to length by downsampling the KL dispersion. WPO simulates the on-policy learning process by adding weights to the optimization objective of DPO. SimPO introduces an optimization objective that does not rely on a reference model, and mitigates the impact of data length by utilizing average likelihood.

**General Training Hyperparameters.** The training hyperparameters are shown in Table.1. Additionally, to ensure the performance of the offline preference optimization algorithms, we set the fitting tuning hyperparameters for all methods. In general, we set $\beta = 0.1$ for DPO, R-DPO, SamPO, WPO and LD-DPO. Specifically, for SimPO, setting $\beta = 2.0$ and $\gamma = 1.0$, for R-DPO, setting $\alpha = 0.05$, and for LD-DPO, we set $\alpha = \{0.1, 0.2, ..., 1.0\}$ to explore its effect on generation length and model performance. Finally, all preference optimization training was conducted on 16 A100-80G GPUs.

| Phase | LR | BS | Epoch | LS | WP |
|-------|-----|-----|-------|--------|-----|
| **SFT** | 2e-5 | 128 | 3 | cosine | 10% |
| **PO** | 5e-7 | 32 | 1 | cosine | 10% |

Table 1: General training hyperparameters settings for SFT phase and preference optimization (PO) phase, including Leaning Rate (LR), Batch Size (BS), Epoch, Learning rate Schedule (LS), Warmup Phase (WP).

## 5 Experimental Results

In this section, we present the main results of our experiments, demonstrating that LD-DPO achieves state-of-the-art (SOTA) performance on both MT-Bench and AlpacaEval for all six settings through effective length control. Building on these results, we further analyze the sensitivity of different

| Method | Llama2-Base (13B) | | | | Llama2-Chat (13B) | | | |
| --- | --- | --- | --- | --- | --- | --- | --- | --- |
| | MT-Bench | | AlpacaEval 2 | | MT-Bench | | AlpacaEval 2 | |
| | Score | Avg. Token | LC (%) | Avg. Token | Score | Avg. Token | LC (%) | Avg. Token |
| SFT | 5.51 | 170 | 6.56 | 220 | 6.35 | 326 | 23.46 | 452 |
| DPO | 5.67 | 191 | 6.70 | 266 | 6.33 | 368 | 25.52 | 487 |
| R-DPO | 5.45 | 150 | 7.64 | 198 | 6.32 | 346 | 26.27 | 461 |
| SimPO | 5.45 | 180 | 7.31 | 246 | 6.40 | 351 | 26.38 | 471 |
| WPO | 5.76 | 185 | 9.65 | 244 | 6.40 | 401 | 26.81 | 486 |
| SamPO | 5.78 | 183 | 8.80 | 259 | 6.21 | 390 | 26.09 | 484 |
| LD-DPO | **5.83** | 154 | **10.37** | 208 | **6.55** | 329 | **28.20** | 449 |

| Method | Llama3-Base (8B) | | | | Llama3-Instruct (8B) | | | |
| --- | --- | --- | --- | --- | --- | --- | --- | --- |
| | MT-Bench | | AlpacaEval 2 | | MT-Bench | | AlpacaEval 2 | |
| | Score | Avg. Token | LC (%) | Avg. Token | Score | Avg. Token | LC (%) | Avg. Token |
| SFT | 6.08 | 156 | 8.40 | 167 | 7.36 | 255 | 38.28 | 326 |
| DPO | 6.38 | 178 | 12.58 | 235 | 7.61 | 323 | 40.21 | 393 |
| R-DPO | 6.18 | 137 | 12.15 | 155 | 7.54 | 248 | 41.07 | 318 |
| SimPO | 6.24 | 142 | 9.96 | 194 | 7.36 | 266 | 39.14 | 374 |
| WPO | 6.42 | 179 | 12.99 | 226 | 7.60 | 320 | 39.77 | 386 |
| SamPO | 6.12 | 162 | 14.62 | 200 | 7.50 | 294 | 40.77 | 368 |
| LD-DPO | **6.45** | 153 | **16.82** | 144 | **7.74** | 247 | **44.00** | 308 |

| Method | Qwen2-Base (7B) | | | | Qwen2-Instruct (7B) | | | |
| --- | --- | --- | --- | --- | --- | --- | --- | --- |
| | MT-Bench | | AlpacaEval 2 | | MT-Bench | | AlpacaEval 2 | |
| | Score | Avg. Token | LC (%) | Avg. Token | Score | Avg. Token | LC (%) | Avg. Token |
| SFT | 6.30 | 160 | 7.62 | 173 | 7.95 | 359 | 34.09 | 373 |
| DPO | 6.73 | 181 | 10.20 | 204 | 7.79 | 321 | 35.63 | 437 |
| R-DPO | 6.16 | 137 | 8.79 | 168 | 7.94 | 314 | 38.85 | 365 |
| SimPO | 6.61 | 154 | 12.08 | 181 | 7.88 | 352 | 35.10 | 430 |
| WPO | 6.71 | 167 | 11.02 | 193 | 7.72 | 361 | 37.53 | 433 |
| SamPO | 6.79 | 180 | 10.89 | 187 | 7.78 | 343 | 37.05 | 399 |
| LD-DPO | **6.80** | 163 | **12.14** | 155 | **8.03** | 303 | **40.88** | 356 |

Table 2: MT-Bench and AlpacaEval 2 results under six model settings. LC-winrate denotes length-controlled win rate against the baseline model (GPT-4-1106-preview), which can mitigate the length preference of the judge model (GPT-4-turbo-0409) compared to the raw win rate. Avg.Token denotes the average length of the model's answers.

models to data length. Additionally, our findings show that our method significantly enhances the model's reasoning ability, with relevant results presented in Appendix C. Finally, we conduct ablation studys and hyperparameter analysis.

## 5.1 MAIN RESULTS

As shown in Table.2, LD-DPO exhibits significant improvements in both MT-Bench and AlpacaEval 2 compared to all other baselines. In addition, the average response length is reduced by 7.8% to 37.9% relative to DPO, suggesting higher quality and more concise model outputs after LD-DPO.

In the Base setting, we observe that the overall model performance is suboptimal, with responses tending to be shorter. This phenomenon may be attributed to the model's performance not being fully realized during the SFT phase. Conversely, in the Instruct setting, the model demonstrates greater competence and generates much longer responses than the base model, due to extensive SFT and RLHF conducted by their publishers. However, in both settings, it is clear that DPO consistently encourages the model to produce more verbose outputs, with experiments showing an increase ranging from 10% to 40%, while LD-DPO can significantly alleviate this issue. This

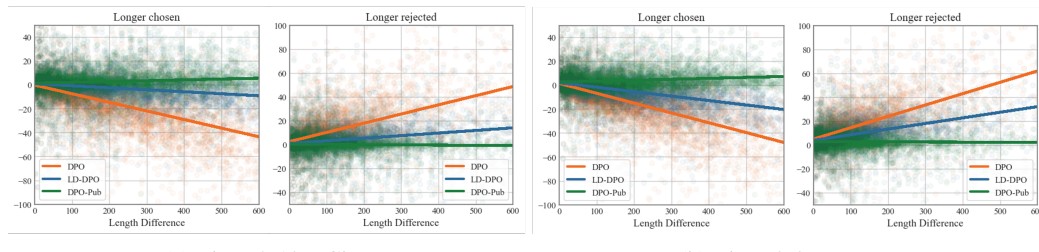

(a) Llama2-13B-Chat                    (b) Llama3-8B-Instruct

Figure 3: Exploring the relationship between predicted probability difference $\log \pi_\theta(y_w|x) - \log \pi_\theta(y_l|x)$ and data length difference under different settings: (a) Llama2-13B-Chat; (b) Llama3-8B-Instruct. In each subplot, the left image represents data where the *chosen* is longer, and the right image represents data where the *rejected* is longer. DPO-Pub indicates that $\alpha = 0$ in LD-DPO. The images depict the true distribution on the UltraFeedback dataset during training.

verbosity can potentially impair model performance, as we will illustrate with several case studies in Appendix D.

As illustrated in Table.2, we examine the average response length and LC-winrate (which may vary from public results due to different judge model) of the models on AlpacaEval 2 under six different settings. Our method achieves the state-of-the-art (SOTA) performance in LC-winrate across all settings. When comparing the three length control methods (R-DPO, SimPO, and SamPO), We find that R-DPO demonstrates superior length control under the Base setting, but its overall performance is suboptimal in terms of LC-winrate. SimPO does not show consistent performance across different settings, likely due to the absence of a reference model, which impacts its stability. SamPO's performance fluctuates less compared to DPO.

## 5.2 LENGTH SENSITIVITY ANALYSIS OF VARIED MODEL

The models selected for our experiments vary in their capabilities and, consequently, in their *length sensitivity* during the DPO process. Their performance based on the choice of the hyperparameter alpha is as follows:

In the Instruct setting, Llama2-13B achieves optimal performance at $\alpha = 0.3$, Llama3-8B at $\alpha = 0.5$, and Qwen2-7B at $\alpha = 0.6$. In the Base setting, the optimal $\alpha$ values for these models are approximately 0.1 to 0.2 lower compared to the Instruct setting. Due to the similarity in performance between Llama3-8B and Qwen2-7B, we subsequently just conducted an in-depth analysis of the Llama2-13B and Llama3-8B models to explore the performance differences between these two sizes. This selection is representative of varying model capacities.

In Fig.3, we plot the distribution of probability difference during preference optimization for Llama2-13B-Chat (Fig.3a) and Llama3-8B-Instruct (Fig.3b), respectively. The data is differentiated based on the length relationship between the *chosen* and *rejected* responses. We will analyze the training process of the LLMs with different hyperparameter settings:

**Under the DPO setting**, we can clearly observe that the difference in predicted probability $\log \pi_\theta(y_w|x) - \log \pi_\theta(y_l|x)$ is influenced by the length of the data, which reflects the implicit reward and determines the optimization direction in DPO. When the *chosen* is longer, the probability difference is smaller than 0 and continues decrease. Based on our previous theoretical analysis, we can infer that it is easier to optimize in the direction of the *chosen* under these conditions, leading to verbose outputs. Conversely, when the *rejected* is longer, the situation is reversed.

**Under the DPO-Pub setting**, where only the public length portions of the $y_w$ and $y_l$ are considered, we find that the the difference in predicted probability is greater than 0 for a larger proportion of the data. This indicates that the models prefer outputting $y_w$, which suggests that both types of models possess sizable base capabilities, with Llama3-8B-Instruct being stronger than Llama2-13B-Chat. Additionally, compared to the DPO setting, the average predicted probability difference of Llama2-13B-Chat increases (decreases) by 19.66 (14.01) and Llama3-8B-Instruct increases (decreases) by 18.37 (13.68), indicating that the former is more significantly affected by the data length.

**Under the LD-DPO setting**, the fact that LD-DPO cannot achieve optimal results in the extreme case of $\alpha = 0$ suggests that longer responses are necessary. This is because additional text can convey more human-like preferences. Furthermore, compared to Llama2-13B-Chat, Llama3-8B-Instruct is more powerful and can capture more human-like preferences from the text. This capability can mitigate the negative effects of response length, indicating that setting $\alpha$ to an extreme value may not be appropriate.

From the above anaylsis, we know that $\alpha$ is actually the result of a compromise to achieve desensitization of DPO based on model capabilities and to prevent the loss of human-like preferences. In addition, we know that different models have different length sensitivities in the DPO process. We define $\gamma$ as the *length sensitivity coefficient*, where $\gamma = 1 - \alpha_s$ and $\alpha_s$ represents the $\alpha$ that yields the best LD-DPO results. A smaller value of $\gamma$ in more capable models indicates that such models are more likely to capture genuine human preferences rather than being influenced by text length. For example, the *length sensitivity coefficient* of Llama3-8B-Instruct is 0.5, whereas that of Llama2-13B-Chat is 0.7. This suggests that the latter is more sensitive to length during DPO.

## 5.3 ABLATION STUDY

To verify the effect of the relative lengths of $y_w$(*chosen*) and $y_l$(*rejected*) in the training data on LD-DPO, we constructed ablation experiments: The length decoupling strategy, as shown in Eq.12, was applied to $y_w$ and $y_l$ separately or simultaneously.

As shown in Table.3, the performance of the LLMs on MT-Bench, AlpacaEval 2 and Arena-Hard, as well as the length control effect, is inferior to that of LD-DPO under both the *chosen* and *rejected* setups. This suggests that length decoupling is necessary for both *chosen* and *rejected*. For more detail, we have:

- If $y_w$ is longer: According to the previous analysis, it is known that DPO is more inclined to optimize in the direction of $y_w$ under the effect of length bias, which results in redundant output. At this point, length decoupling for $y_w$ can alleviate this tendency.

- If $y_l$ is longer: DPO tends to block the output of $y_l$. In this case, the decoupling of the length of $y_l$ may redirect the optimization towards a shorter $y_w$, thus reducing redundancy and improving the quality of the model's answers.

Therefore, our length desensitization strategy can encourage the model to identify and leverage more human-like preference gaps in the data samples, rather than optimizing the training process based on superficial length differences.

| Method | MT-Bench | | AlpacaEval 2 | | Arena-Hard | |
|---|---|---|---|---|---|---|
| | Score | Avg. Token | LC (%) | Avg. Token | Win-Rate | Avg. Token |
| SFT | 7.36 | 255 | 38.28 | 326 | 24.3 | 470 |
| DPO | 7.61 | 323 | 40.21 | 393 | 27.6 | 560 |
| *chosen* | 7.67 | 302($\downarrow$21) | 43.39 | 352($\downarrow$41) | 27.2 | 526($\downarrow$34) |
| *rejected* | 7.62 | 283($\downarrow$40) | 42.23 | 351($\downarrow$42) | 27.9 | 523($\downarrow$37) |
| LD-DPO | **7.74** | **247($\downarrow$76)** | **44.00** | **308($\downarrow$85)** | **28.0** | **485($\downarrow$75)** |

Table 3: Ablation study on Llama3-8B-Instruct: *chosen* denotes length decoupling (Eq.12) applied only to $y_w$, *rejected* denotes only to $y_l$, and $\downarrow$tokens denotes the Avg. Token drop compared to DPO.

## 5.4 HYPERPARAMETER ANALYSIS

In this subsection, we verify the effect of the hyperparameter $\alpha$ on the performance of LD-DPO. We present the experimental results of Llama3-8B-Instruct. In all experiments, we set $\beta = 0.1$, though we encourage researchers to explore the effects of different $\alpha$ selected at various $\beta$ settings.

When $\alpha = 1$, LD-DPO is equivalent to DPO. As $\alpha$ gradually decreases, the degree of length decoupling increases. At this point, as shown in Fig.4, the sensitivity of the training process to length begins to decline, resulting in a subsequent reduction in the average output length.

We find that the decrease in average response length is pronounced as the parameter $\alpha$ decreases from 1 to 0.5. However, this tendency becomes less significant as $\alpha$ further decreases from 0.5 to 0. This phenomenon can likely be attributed to the fact that verbosity preference decoupling is effectively complete when $\alpha$ reaches 0.5.

In terms of the performance on AlpacaEval 2 and MT-Bench, the trend is to first increase and then decrease as $\alpha$ decreases, with the best performance observed at $\alpha = 0.5$. This indicates that when $\alpha$ is too large, the model's performance is constrained by DPO's *length sensitivity*, resulting in verbose, poor-quality content. Conversely, when $\alpha$ is too small, excessive length decoupling leads to a loss of human-like preferences in the text, thereby reducing the optimization effectiveness.

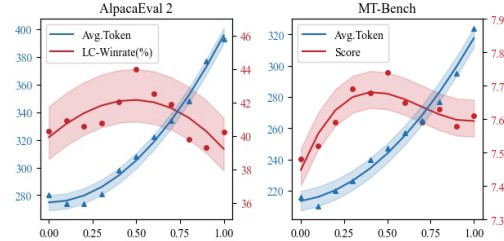

Figure 4: Hyperparametric analysis on $\alpha$ with Llama3-8B-Instruct on AlpacaEval 2(left) and MT-Bench(right).

## 6 RELATED WORKS

### 6.1 OFFLINE PREFERENCE OPTIMIZATION

In practice, traditional RLHF paradigms are more complex in terms of coding and hyperparameter tuning, requiring four models simultaneously, which makes them more resource-intensive and difficult to train stably. Due to the lack of online reward models, DPO needs to construct artificial preference datasets in advance, and many works have proposed different data construction strategies to enable the model to better learn human preferences (An et al., 2023; Gallego, 2024; Khaki et al., 2024). Meanwhile, another research direction is to improve the preference optimization objective, including the necessity of the reference model, the selection of the reward fitting function, and the adjustment of the update weights, and derive a variety of offline optimization strategies, including ORPO (Hong et al., 2024), KTO (Ethayarajh et al., 2024), NCA (Chen et al., 2024b), IPO (Azar et al., 2024), WPO (Zhou et al., 2024), etc.

### 6.2 LENGTH CONTROL STRATEGY

Recent research has shown that DPO may lead to biased results, such as models producing lengthy outputs, which affects the model's ability to follow instructions and reasoning. To address this problem, Park et al. (2024) proposed R-DPO, which suppresses the model from producing excessively long answers by introducing a rule term, which is a intuitive scheme followed by several algorithms (Zhou et al., 2024; Chen et al., 2024a). Meng et al. (2024) proposed an optimization strategy that does not rely on the reference model, called SimPO, which uses length-normalized rewards to prevent the model from generating excessively long but low-quality answers. Lu et al. (2024) argued that the phenomenon of lengthy outputs in DPO is due to the overestimation or underestimation of implicit rewards caused by the length of the training data. Based on this, they proposed SamPO, which mitigates the length bias by down-sampling the KL divergence to ensure that implicit rewards are not affected by length.

## 7 CONCLUSION

In this work, we propose for the first time that the optimization process of DPO is length-sensitive and provide a theoretical proof. Based on this, we design a length-desensitization algorithm based on DPO: LD-DPO, which achieves length desensitization by reparameterizing the likelihood to decouple verbosity preferences from complete information while preserving human-like preferences. Through extensive experimental analysis, LD-DPO consistently outperforms existing algorithms in various training settings, achieving performance improvements with a 10-40% reduction in output length, especially in reasoning ability. This suggests that previous optimization algorithms have overemphasized length at the expense of quality, validating the value of our work. Furthermore, we perform a comparative analysis of *length sensitivity* across models with different capabilities, which may provide new insights into the preference optimization process.

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

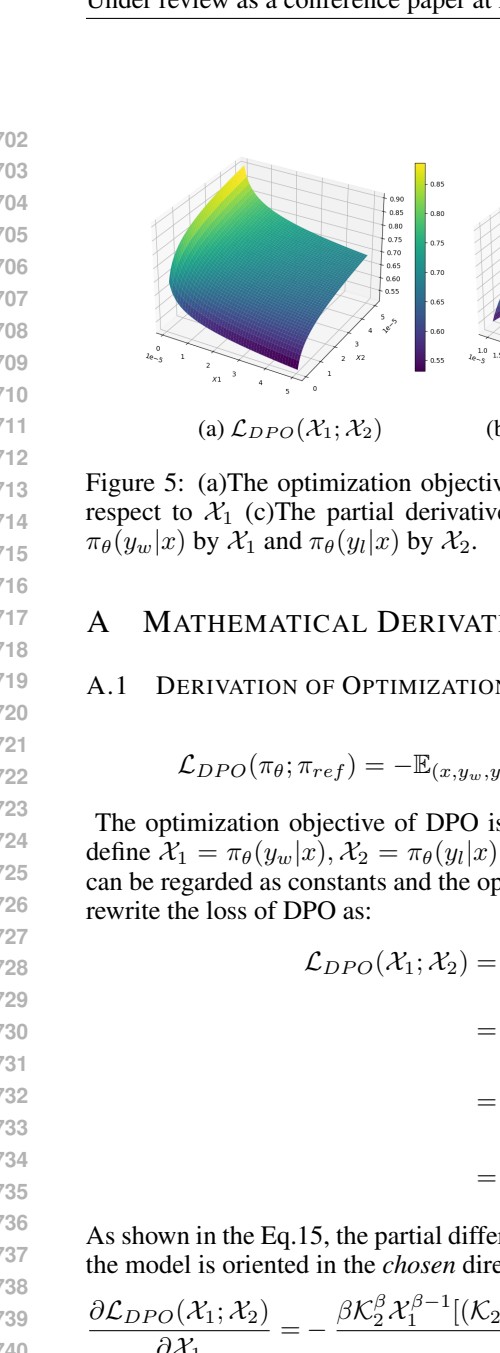

(a) $\mathcal{L}_{DPO}(\mathcal{X}_1; \mathcal{X}_2)$      (b) $\partial \mathcal{L}_{DPO}(\mathcal{X}_1; \mathcal{X}_2)/\partial \mathcal{X}_1$      (c) $\partial \mathcal{L}_{DPO}(\mathcal{X}_1; \mathcal{X}_2)/\partial \mathcal{X}_2$

Figure 5: (a)The optimization objective of DPO (b)The partial derivative of $\mathcal{L}_{DPO}(\mathcal{X}_1; \mathcal{X}_2)$ with respect to $\mathcal{X}_1$ (c)The partial derivative of $\mathcal{L}_{DPO}(\mathcal{X}_1; \mathcal{X}_2)$ with respect to $\mathcal{X}_2$, where we denote $\pi_\theta(y_w|x)$ by $\mathcal{X}_1$ and $\pi_\theta(y_l|x)$ by $\mathcal{X}_2$.

# A   MATHEMATICAL DERIVATIONS

## A.1   DERIVATION OF OPTIMIZATION DIRECTION IN DPO

$$\mathcal{L}_{DPO}(\pi_\theta; \pi_{ref}) = -\mathbb{E}_{(x,y_w,y_l)\sim\mathcal{D}}[\log \sigma(\beta \log \frac{\pi_\theta(y_w|x)}{\pi_{ref}(y_w|x)} - \beta \log \frac{\pi_\theta(y_l|x)}{\pi_{ref}(y_l|x)})]. \tag{13}$$

The optimization objective of DPO is presented in Eq.13 and its image is shown in Fig.5a. We define $\mathcal{X}_1 = \pi_\theta(y_w|x)$, $\mathcal{X}_2 = \pi_\theta(y_l|x)$ and $\mathcal{K}_1 = \pi_{ref}(y_w|x)$, $\mathcal{K}_2 = \pi_{ref}(y_l|x)$, where $\mathcal{K}_1$ and $\mathcal{K}_2$ can be regarded as constants and the optimization process of DPO is only related to $\mathcal{X}_1, \mathcal{X}_2$. We can rewrite the loss of DPO as:

$$\begin{aligned} \mathcal{L}_{DPO}(\mathcal{X}_1; \mathcal{X}_2) &= -\log \sigma(\beta \log \frac{\mathcal{X}_1}{\mathcal{K}_1} - \beta \log \frac{\mathcal{X}_2}{\mathcal{K}_2}) \\ &= -\log \frac{1}{1 + exp\{-\beta \log(\mathcal{K}_2\mathcal{X}_1/\mathcal{K}_1\mathcal{X}_2)\}} \\ &= -\log \frac{1}{1 + (\mathcal{K}_1\mathcal{X}_2/\mathcal{K}_2\mathcal{X}_1)^\beta} \\ &= -\log \frac{(\mathcal{K}_2\mathcal{X}_1)^\beta}{(\mathcal{K}_2\mathcal{X}_1)^\beta + (\mathcal{K}_1\mathcal{X}_2)^\beta}. \end{aligned} \tag{14}$$

As shown in the Eq.15, the partial differentiation of $\mathcal{L}_{DPO}(\mathcal{X}_1; \mathcal{X}_2)$ with respect to $\mathcal{X}_1$ indicates that the model is oriented in the *chosen* direction, the function image is shown in Fig.5b.

$$\begin{aligned} \frac{\partial \mathcal{L}_{DPO}(\mathcal{X}_1; \mathcal{X}_2)}{\partial \mathcal{X}_1} &= -\frac{\beta\mathcal{K}_2^\beta\mathcal{X}_1^{\beta-1}[(\mathcal{K}_2\mathcal{X}_1)^\beta + (\mathcal{K}_1\mathcal{X}_2)^\beta] - (\mathcal{K}_2\mathcal{X}_1)^\beta\beta\mathcal{K}_2^\beta\mathcal{X}_1^{\beta-1}}{[(\mathcal{K}_2\mathcal{X}_1)^\beta + (\mathcal{K}_1\mathcal{X}_2)^\beta]^2} \frac{(\mathcal{K}_2\mathcal{X}_1)^\beta + (\mathcal{K}_1\mathcal{X}_2)^\beta}{(\mathcal{K}_2\mathcal{X}_1)^\beta} \\ &= -\frac{\beta\mathcal{K}_2^\beta\mathcal{X}_1^{\beta-1}(\mathcal{K}_2\mathcal{X}_1)^\beta + \beta\mathcal{K}_2^\beta\mathcal{X}_1^{\beta-1}(\mathcal{K}_1\mathcal{X}_2)^\beta - (\mathcal{K}_2\mathcal{X}_1)^\beta\beta\mathcal{K}_2^\beta\mathcal{X}_1^{\beta-1}}{[(\mathcal{K}_2\mathcal{X}_1)^\beta + (\mathcal{K}_1\mathcal{X}_2)^\beta](\mathcal{K}_2\mathcal{X}_1)^\beta} \\ &= -\frac{\beta\mathcal{K}_2^\beta\mathcal{X}_1^{\beta-1}(\mathcal{K}_1\mathcal{X}_2)^\beta}{[(\mathcal{K}_2\mathcal{X}_1)^\beta + (\mathcal{K}_1\mathcal{X}_2)^\beta](\mathcal{K}_2\mathcal{X}_1)^\beta} \\ &= -\frac{\beta(\mathcal{K}_1\mathcal{X}_2)^\beta}{\mathcal{X}_1[(\mathcal{K}_2\mathcal{X}_1)^\beta + (\mathcal{K}_1\mathcal{X}_2)^\beta]}. \end{aligned} \tag{15}$$

As shown in the Eq.16, the partial differentiation of $\mathcal{L}_{DPO}(\mathcal{X}_1; \mathcal{X}_2)$ with respect to $\mathcal{X}_2$ indicates that the model is oriented in the *rejected* direction, the function image is shown in Fig.5c.

$$\begin{aligned} \frac{\partial \mathcal{L}_{DPO}(\mathcal{X}_1; \mathcal{X}_2)}{\partial \mathcal{X}_2} &= -\frac{-(\mathcal{K}_2\mathcal{X}_1)^\beta\beta\mathcal{K}_1^\beta\mathcal{X}_2^{\beta-1}}{[(\mathcal{K}_2\mathcal{X}_1)^\beta + (\mathcal{K}_1\mathcal{X}_2)^\beta]^2} \frac{(\mathcal{K}_2\mathcal{X}_1)^\beta + (\mathcal{K}_1\mathcal{X}_2)^\beta}{(\mathcal{K}_2\mathcal{X}_1)^\beta} \\ &= \frac{\beta\mathcal{K}_1^\beta\mathcal{X}_2^{\beta-1}}{(\mathcal{K}_2\mathcal{X}_1)^\beta + (\mathcal{K}_1\mathcal{X}_2)^\beta}. \end{aligned} \tag{16}$$

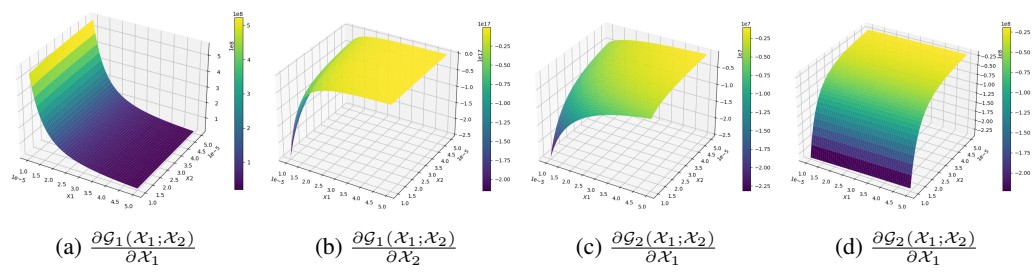

(a) $\frac{\partial \mathcal{G}_1(\mathcal{X}_1;\mathcal{X}_2)}{\partial \mathcal{X}_1}$  (b) $\frac{\partial \mathcal{G}_1(\mathcal{X}_1;\mathcal{X}_2)}{\partial \mathcal{X}_2}$  (c) $\frac{\partial \mathcal{G}_2(\mathcal{X}_1;\mathcal{X}_2)}{\partial \mathcal{X}_1}$  (d) $\frac{\partial \mathcal{G}_2(\mathcal{X}_1;\mathcal{X}_2)}{\partial \mathcal{X}_1}$

Figure 6: (a)The partial derivative of $\mathcal{G}_1(\mathcal{X}_1;\mathcal{X}_2)$ with respect to $\mathcal{X}_1$; (b)The partial derivative of $\mathcal{G}_1(\mathcal{X}_1;\mathcal{X}_2)$ with respect to $\mathcal{X}_2$; (c)The partial derivative of $\mathcal{G}_2(\mathcal{X}_1;\mathcal{X}_2)$ with respect to $\mathcal{X}_1$; (d)The partial derivative of $\mathcal{G}_2(\mathcal{X}_1;\mathcal{X}_2)$ with respect to $\mathcal{X}_2$.

According to the Eq.15 and Eq.16, we can obtain Eq.17, which implies that the value of the gradient of the DPO loss function in the preference direction, compared to the non-preference direction, is inversely proportional to its prediction probability.

$$\left| \frac{\partial \mathcal{L}_{DPO}(\mathcal{X}_1;\mathcal{X}_2)}{\partial \mathcal{X}_1} \Big/ \frac{\partial \mathcal{L}_{DPO}(\mathcal{X}_1;\mathcal{X}_2)}{\partial \mathcal{X}_2} \right| = \frac{\mathcal{X}_2}{\mathcal{X}_1} = \frac{\pi_\theta(y_l|x)}{\pi_\theta(y_w|x)}. \tag{17}$$

Next, we analyze the relationship between the partial gradient values $\frac{\partial \mathcal{L}_{DPO}(\mathcal{X}_1;\mathcal{X}_2)}{\partial \mathcal{X}_1}$, $\frac{\partial \mathcal{L}_{DPO}(\mathcal{X}_1;\mathcal{X}_2)}{\partial \mathcal{X}_2}$ and $\mathcal{X}_1, \mathcal{X}_2$, and we denote $\frac{\partial \mathcal{L}_{DPO}(\mathcal{X}_1;\mathcal{X}_2)}{\partial \mathcal{X}_1}$ as $\mathcal{G}_1(\mathcal{X}_1;\mathcal{X}_2)$ and $\frac{\partial \mathcal{L}_{DPO}(\mathcal{X}_1;\mathcal{X}_2)}{\partial \mathcal{X}_2}$ as $\mathcal{G}_2(\mathcal{X}_1;\mathcal{X}_2)$. In addition, it is known that $\mathcal{X}_1, \mathcal{X}_2, \mathcal{K}_1, \mathcal{K}_2$ all represent likelihood, each taking values in the range $(0,1)$, and $\beta$ is a hyperparameter that is range from $(0,1)$ in DPO. We solve for the partial derivatives with respect to $\mathcal{X}_1$ and $\mathcal{X}_2$ for $\mathcal{G}_1(\mathcal{X}_1;\mathcal{X}_2)$ and $\mathcal{G}_2(\mathcal{X}_1;\mathcal{X}_2)$ between Eq.18 and Eq.21, and their images are shown in Fig. 6.

$$\begin{aligned} \frac{\partial \mathcal{G}_1(\mathcal{X}_1;\mathcal{X}_2)}{\partial \mathcal{X}_1} &= \frac{\beta(\mathcal{K}_1\mathcal{X}_2)^\beta[(\mathcal{K}_2\mathcal{X}_1)^\beta + (\mathcal{K}_1\mathcal{X}_2)^\beta + \mathcal{X}_1\beta\mathcal{K}_2^\beta\mathcal{X}_1^{\beta-1}]}{\mathcal{X}_1^2[(\mathcal{K}_2\mathcal{X}_1)^\beta + (\mathcal{K}_1\mathcal{X}_2)^\beta]^2} \\ &= \frac{\beta(\mathcal{K}_1\mathcal{X}_2)^{2\beta} + \beta(\beta+1)(\mathcal{K}_1\mathcal{X}_2)^\beta(\mathcal{K}_2\mathcal{X}_1)^\beta}{\mathcal{X}_1^2[(\mathcal{K}_2\mathcal{X}_1)^\beta + (\mathcal{K}_1\mathcal{X}_2)^\beta]^2} > 0. \end{aligned} \tag{18}$$

$$\begin{aligned} \frac{\partial \mathcal{G}_1(\mathcal{X}_1;\mathcal{X}_2)}{\partial \mathcal{X}_2} &= -\frac{\beta^2\mathcal{K}_1^\beta\mathcal{X}_2^{\beta-1}\mathcal{X}_1[(\mathcal{K}_2\mathcal{X}_1)^\beta + (\mathcal{K}_1\mathcal{X}_2)^\beta] - \beta(\mathcal{K}_1\mathcal{X}_2)^\beta\beta\mathcal{K}_1^\beta\mathcal{X}_1\mathcal{X}_2^{\beta-1}}{\mathcal{X}_1^2[(\mathcal{K}_2\mathcal{X}_1)^\beta + (\mathcal{K}_1\mathcal{X}_2)^\beta]^2} \\ &= -\frac{\beta^2\mathcal{K}_1^\beta\mathcal{X}_2^{\beta-1}\mathcal{X}_1(\mathcal{K}_2\mathcal{X}_1)^\beta}{\mathcal{X}_1^2[(\mathcal{K}_2\mathcal{X}_1)^\beta + (\mathcal{K}_1\mathcal{X}_2)^\beta]^2} < 0. \end{aligned} \tag{19}$$

$$\begin{aligned} \frac{\partial \mathcal{G}_2(\mathcal{X}_1;\mathcal{X}_2)}{\partial \mathcal{X}_1} &= -\frac{\beta\mathcal{K}_1^\beta\mathcal{X}_2^{\beta-1}\beta\mathcal{K}_2^\beta\mathcal{X}_1^{\beta-1}}{[(\mathcal{K}_2\mathcal{X}_1)^\beta + (\mathcal{K}_1\mathcal{X}_2)^\beta]^2} \\ &= -\frac{\beta^2\mathcal{K}_1^\beta\mathcal{X}_2^{\beta-1}\mathcal{K}_2^\beta\mathcal{X}_1^{\beta-1}}{[(\mathcal{K}_2\mathcal{X}_1)^\beta + (\mathcal{K}_1\mathcal{X}_2)^\beta]^2} < 0. \end{aligned} \tag{20}$$

$$\begin{aligned} \frac{\partial \mathcal{G}_2(\mathcal{X}_1;\mathcal{X}_2)}{\partial \mathcal{X}_2} &= \frac{\beta(\beta-1)\mathcal{K}_1^\beta\mathcal{X}_2^{\beta-2}[(\mathcal{K}_2\mathcal{X}_1)^\beta + (\mathcal{K}_1\mathcal{X}_2)^\beta] - \beta\mathcal{K}_1^\beta\mathcal{X}_2^{\beta-1}\beta\mathcal{K}_1^\beta\mathcal{X}_2^{\beta-1}}{[(\mathcal{K}_2\mathcal{X}_1)^\beta + (\mathcal{K}_1\mathcal{X}_2)^\beta]^2} \\ &= \frac{\beta(\beta-1)\mathcal{K}_1^\beta\mathcal{X}_2^{\beta-2}\mathcal{K}_2^\beta\mathcal{X}_1^\beta - \beta\mathcal{K}_1^{2\beta}\mathcal{K}_2^{2\beta-2}}{[(\mathcal{K}_2\mathcal{X}_1)^\beta + (\mathcal{K}_1\mathcal{X}_2)^\beta]^2} < 0. \end{aligned} \tag{21}$$

Based on the analysis of the aforementioned function trend, we can draw the following conclusion: When $\mathcal{X}_1$ decreases, both $\left| \frac{\partial \mathcal{L}_{DPO}(\mathcal{X}_1;\mathcal{X}_2)}{\partial \mathcal{X}_1} \right|$ and $\left| \frac{\partial \mathcal{L}_{DPO}(\mathcal{X}_1;\mathcal{X}_2)}{\partial \mathcal{X}_2} \right|$ increase. When $\mathcal{X}_2$ decreases,

$\left|\frac{\partial \mathcal{L}_{DPO}(\mathcal{X}_1;\mathcal{X}_2)}{\partial \mathcal{X}_1}\right|$ decreases and $\left|\frac{\partial \mathcal{L}_{DPO}(\mathcal{X}_1;\mathcal{X}_2)}{\partial \mathcal{X}_2}\right|$ increases. According to the analysis in Subsection 3.1, when the training data is more extensive, the parameters will be updated at a faster rate, thereby exacerbating the sensitivity of the DPO.

## A.2 DERIVATION OF THE MODIFIED LIKELIHOOD IN LD-DPO

$$
\begin{aligned}
\hat{\pi}_\theta(y|x) &= \prod_{i=1}^{l_p} p(y_i|x, y_{<i}) \prod_{i=l_p+1}^{l} p^\alpha(y_i|x, y_{<i}) \\
&= \prod_{i=1}^{l_p} p^\alpha(y_i|x, y_{<i}) p^{1-\alpha}(y_i|x, y_{<i}) \prod_{i=l_p+1}^{l} p^\alpha(y_i|x, y_{<i}) \\
&= \prod_{i=1}^{l_p} p^\alpha(y_i|x, y_{<i}) \prod_{i=l_p+1}^{l} p^\alpha(y_i|x, y_{<i}) \prod_{i=1}^{l_p} p^{1-\alpha}(y_i|x, y_{<i}) \\
&= \prod_{i=1}^{l} p^\alpha(y_i|x, y_{<i}) \prod_{i=1}^{l_p} p^{1-\alpha}(y_i|x, y_{<i}).
\end{aligned}
\tag{22}
$$

## B ADDITIONAL EXPERIMENTAL RESULTS

### B.1 ARENA-HARD RESULTS

To comprehensively validate the effectiveness of LD-DPO, we conduct experimental evaluations on Arena-Hard(Li et al., 2024), an enhanced version of MT-Bench. Arena-Hard consists of 500 well-defined technical problem-solving queries and presents more complex problem scenarios compared to MT-Bench and AlpacaEval, thereby posing a greater challenge to model performance. We select the Instruct model, known for its superior modeling capabilities, as the initial model to compare the performance differences between LD-DPO and five other offline optimization algorithms.

As shown in Table.4, the overall performance of LD-DPO surpasses the five offline preference optimization strategies, including DPO, with a significant decrease in average response length compared to DPO. For Llama2-13B-Chat and Llama3-8B-Instruct, LD-DPO performs suboptimally; however, the average response length is more than 10% shorter compared to the SOTA algorithm. In the case of Qwen2-Instruct, LD-DPO achieves the highest Win rate of **31.2%** (exceeding DPO by 7.8%), and the average response length is 10% shorter. Notably, GPT-4-turbo-0409, the judge model, exhibits a clear length preference, favoring longer responses when calculating win rate. Despite this, LD-DPO achieves high win rates with shorter responses, which strongly indicates its effectiveness in aligning with human-like preferences.

| Method | Llama2-Chat (13B) | | Llama3-Instruct (8B) | | Qwen2-Instruct (7B) | | Avg.WR(%) |
|---|---|---|---|---|---|---|---|
| | WR(%) | Avg. Token | WR(%) | Avg. Token | WR(%) | Avg. Token | |
| SFT | 9.6 | 635 | 24.3 | 470 | 22.9 | 533 | 18.9 |
| DPO | **10.2** | 661 | 27.6 | 560 | 23.4 | 576 | 20.4 |
| R-DPO | 9.8 | 607 | 27.1 | 455 | 29.1 | 538 | 22.0 |
| SimPO | 10.1 | 620 | 23.9 | 495 | 26.0 | 603 | 20.0 |
| WPO | 9.6 | 665 | **28.8** | 553 | 24.6 | 579 | 21.0 |
| SamPO | 9.7 | 637 | 28.2 | 533 | 25.6 | 580 | 21.2 |
| LD-DPO | 10.1 | 603 | 28.0 | 485 | **31.2** | 516 | **23.1** |

Table 4: Arena-Hard results under Instruct model setting: WR denotes the win rate against the baseline model (GPT-4-0314) as judged by GPT-4-turbo-0409. Avg.Token denotes the average length of the model's answers. Avg.WR denotes the average win rate of three models.

| Method | GSM8K | BBH | WinoGrande | CSQA | ARC | MMLU | HellaSwag | ProofWriter | Average |
|---|---|---|---|---|---|---|---|---|---|
| | | | | Llama2-13B-Base | | | | | |
| SFT | 34.18 | 37.61 | 53.08 | 69.37 | 73.51 | 50.79 | 36.38 | 48.69 | 50.45 |
| DPO | 35.64 | 37.95 | 53.20 | 69.33 | 73.41 | 50.68 | 37.37 | 47.89 | 50.68 |
| R-DPO | 32.53 | 37.68 | 53.12 | 68.84 | 73.03 | 50.52 | 38.13 | 48.19 | 50.26 |
| SimPO | 32.68 | 36.70 | 52.29 | 66.75 | 72.88 | 50.53 | 36.26 | 47.92 | 49.50 |
| WPO | 35.03 | 37.26 | 53.12 | 69.29 | 73.35 | 50.57 | 36.95 | 47.67 | 50.40 |
| SamPO | 34.33 | 37.22 | 53.20 | 69.08 | 73.30 | 50.39 | 37.15 | 48.50 | 50.40 |
| LD-DPO | 35.07 | 37.97 | 53.16 | 69.16 | 73.55 | 50.91 | 39.10 | 48.83 | 50.97 |
| | | | | Llama2-13B-Chat | | | | | |
| SFT | 43.96 | 44.69 | 50.99 | 64.05 | 73.22 | 55.28 | 49.00 | 47.08 | 53.53 |
| DPO | 44.11 | 44.98 | 51.50 | 64.91 | 73.14 | 55.40 | 49.20 | 47.72 | 53.87 |
| R-DPO | 43.73 | 44.35 | 51.78 | 65.27 | 73.05 | 54.68 | 49.45 | 47.42 | 53.72 |
| SimPO | 44.02 | 44.91 | 50.95 | 64.21 | 71.53 | 54.65 | 48.90 | 47.50 | 53.33 |
| WPO | 43.65 | 44.64 | 51.42 | 64.78 | 73.19 | 55.14 | 48.81 | 47.56 | 53.65 |
| SamPO | 44.73 | 44.57 | 51.46 | 64.74 | 73.00 | 55.36 | 48.95 | 47.94 | 53.84 |
| LD-DPO | 43.80 | 44.70 | 51.78 | 65.32 | 73.09 | 55.21 | 49.64 | 47.89 | 53.93 |
| | | | | Llama3-8B-Base | | | | | |
| SFT | 56.27 | 45.53 | 54.03 | 70.68 | 83.10 | 61.61 | 43.66 | 52.75 | 58.45 |
| DPO | 56.66 | 46.49 | 54.70 | 71.70 | 83.23 | 62.29 | 46.42 | 48.89 | 58.80 |
| R-DPO | 53.58 | 45.67 | 54.30 | 71.25 | 83.35 | 62.46 | 48.70 | 50.61 | 58.74 |
| SimPO | 54.43 | 45.94 | 54.18 | 71.50 | 83.53 | 62.17 | 46.70 | 51.56 | 58.75 |
| WPO | 56.74 | 46.27 | 54.06 | 71.74 | 83.37 | 62.35 | 46.32 | 50.50 | 58.92 |
| SamPO | 57.74 | 45.69 | 54.38 | 71.66 | 83.44 | 62.55 | 46.75 | 49.61 | 58.98 |
| LD-DPO | 58.12 | 46.39 | 54.54 | 71.62 | 83.56 | 62.58 | 49.33 | 51.39 | 59.69 |
| | | | | Llama3-8B-Instruct | | | | | |
| SFT | 82.60 | 62.07 | 61.33 | 76.17 | 87.87 | 69.40 | 58.70 | 55.19 | 69.17 |
| DPO | 82.68 | 61.24 | 60.89 | 75.76 | 87.65 | 67.65 | 58.82 | 56.14 | 68.85 |
| R-DPO | 83.53 | 60.21 | 59.75 | 75.35 | 86.56 | 66.68 | 58.44 | 57.19 | 68.47 |
| SimPO | 83.22 | 58.54 | 59.94 | 75.76 | 86.60 | 66.87 | 59.55 | 53.69 | 68.02 |
| WPO | 82.37 | 61.18 | 61.52 | 76.78 | 87.76 | 70.01 | 59.63 | 56.47 | 69.47 |
| SamPO | 83.83 | 61.73 | 60.34 | 75.88 | 87.38 | 67.76 | 58.87 | 57.42 | 69.15 |
| LD-DPO | 83.76 | 62.10 | 60.97 | 76.41 | 87.41 | 67.79 | 59.66 | 58.72 | 69.61 |
| | | | | Qwen2-7B-Base | | | | | |
| SFT | 82.53 | 50.30 | 62.98 | 76.41 | 89.48 | 69.59 | 56.12 | 54.50 | 67.74 |
| DPO | 82.91 | 50.83 | 62.15 | 76.00 | 89.75 | 69.05 | 55.22 | 51.64 | 67.20 |
| R-DPO | 83.53 | 49.52 | 62.08 | 76.00 | 89.53 | 68.99 | 55.62 | 51.58 | 67.11 |
| SimPO | 82.72 | 48.22 | 62.67 | 76.45 | 89.52 | 68.80 | 57.07 | 53.42 | 67.36 |
| WPO | 84.30 | 49.86 | 62.12 | 75.76 | 89.74 | 68.87 | 55.24 | 54.06 | 67.41 |
| SamPO | 83.37 | 49.92 | 62.15 | 75.96 | 89.66 | 68.95 | 55.26 | 52.97 | 67.28 |
| LD-DPO | 84.06 | 50.46 | 62.47 | 76.29 | 89.74 | 69.12 | 56.37 | 54.61 | 67.88 |
| | | | | Qwen2-7B-Instruct | | | | | |
| SFT | 88.61 | 57.82 | 66.02 | 78.54 | 89.77 | 71.06 | 70.43 | 58.14 | 72.55 |
| DPO | 87.84 | 58.27 | 65.79 | 78.42 | 89.74 | 71.39 | 69.73 | 59.11 | 72.54 |
| R-DPO | 88.99 | 56.99 | 65.98 | 78.05 | 89.69 | 70.73 | 71.06 | 58.31 | 72.48 |
| SimPO | 88.14 | 59.32 | 66.30 | 78.13 | 89.83 | 71.08 | 71.75 | 58.03 | 72.83 |
| WPO | 87.84 | 58.72 | 65.90 | 78.21 | 89.80 | 71.09 | 69.62 | 57.97 | 72.40 |
| SamPO | 87.84 | 58.27 | 65.79 | 78.42 | 89.83 | 71.39 | 69.73 | 59.58 | 72.61 |
| LD-DPO | 88.99 | 58.07 | 66.34 | 78.46 | 89.75 | 70.84 | 71.16 | 59.69 | 72.90 |

Table 5: Results on downstream tasks on the Huggingface OpenLLM Leaderboard.

## B.2 RESULTS ON DOWNSTREAM TASKS

We evaluate the performances of LD-DPO and all baselines on various tasks on OpenLLM leaderboard (Beeching et al., 2023), including MMLU (Hendrycks et al., 2021), ARC (Clark et al., 2018), BBH (Suzgun et al., 2023), GSM8K (Cobbe et al., 2021), CommonsenseQA (Talmor et al., 2019), WinoGrande (Sakaguchi et al., 2021), HellaSwag (Zellers et al., 2019) and ProofWriter (Tafjord et al., 2021). The results are shown in Table 5, from where we find that:

- LD-DPO outperforms SFT, DPO, and all other baselines on the average score across all benchmarks in all settings.

- Preference optimization, whose goal is to align LLMs with human preference, may not significantly improve the performance on all downstream tasks.

- All preference optimization methods perform comparably to SFT model on MMLU and CommonsenseQA(CSQA), with slight fluctuations, showing that knowledge is maintained during the preference optimization stage.

- Compared to SFT, DPO, and other baselines, LD-DPO significantly improves the performance on HellaSwag and ProofWriter, indicating that LD-DPO can enhance the reasoning capability of LLMs.

| Method | Writing | Roleplay | Reasoning | Math | Coding | Extraction | STEM | Humanities | Average |
|--------|---------|----------|-----------|------|--------|------------|------|------------|---------|
| **Llama2-13B-Base** | | | | | | | | | |
| SFT | 8.15 | 6.20 | 4.35 | 1.65 | 2.95 | 6.75 | 5.90 | 8.15 | 5.51 |
| DPO | 7.55 | 7.05 | 4.95 | 1.70 | 3.00 | 6.55 | 6.30 | 8.30 | 5.67 |
| R-DPO | 8.05 | 5.90 | 3.80 | 2.10 | 2.90 | 6.35 | 6.10 | 8.40 | 5.45 |
| SimPO | 7.95 | 6.30 | 4.60 | 1.70 | 3.05 | 6.60 | 5.75 | 8.05 | 5.50 |
| WPO | 7.75 | 6.70 | 4.60 | 1.80 | 2.85 | 7.55 | 6.30 | 8.55 | 5.76 |
| SamPO | 7.85 | 6.55 | 4.75 | 1.35 | 3.05 | 8.05 | 6.40 | 8.25 | 5.78 |
| LD-DPO | 7.80 | 6.50 | 4.75 | 1.60 | 3.65 | 7.40 | 6.55 | 8.45 | 5.83 |
| **Llama2-13B-Chat** | | | | | | | | | |
| SFT | 8.45 | 7.20 | 5.35 | 3.30 | 2.75 | 7.30 | 7.50 | 9.00 | 6.35 |
| DPO | 7.90 | 7.50 | 5.40 | 2.95 | 3.15 | 7.05 | 7.30 | 9.40 | 6.33 |
| R-DPO | 8.75 | 7.05 | 5.65 | 3.05 | 3.00 | 7.25 | 6.80 | 9.05 | 6.32 |
| SimPO | 8.90 | 7.25 | 5.55 | 3.05 | 3.50 | 6.75 | 7.15 | 9.05 | 6.40 |
| WPO | 8.50 | 6.65 | 5.60 | 2.60 | 3.05 | 7.95 | 7.60 | 9.25 | 6.40 |
| SamPO | 8.20 | 7.05 | 5.45 | 2.70 | 3.00 | 7.85 | 6.55 | 8.90 | 6.21 |
| LD-DPO | 8.60 | 7.30 | 6.05 | 3.20 | 3.25 | 7.20 | 7.65 | 9.15 | 6.55 |
| **Llama3-8B-Base** | | | | | | | | | |
| SFT | 7.80 | 6.25 | 3.90 | 3.05 | 4.25 | 8.40 | 6.55 | 8.50 | 6.08 |
| DPO | 7.95 | 6.80 | 4.05 | 3.20 | 4.45 | 8.75 | 7.25 | 8.65 | 6.38 |
| R-DPO | 8.30 | 6.30 | 4.00 | 2.65 | 4.10 | 8.30 | 7.45 | 8.35 | 6.18 |
| SimPO | 8.10 | 6.30 | 4.40 | 3.05 | 4.35 | 8.35 | 7.15 | 8.20 | 6.24 |
| WPO | 8.45 | 6.60 | 4.10 | 3.45 | 4.60 | 8.50 | 7.10 | 8.50 | 6.42 |
| SamPO | 8.15 | 6.40 | 4.40 | 2.75 | 4.30 | 8.50 | 6.40 | 8.20 | 6.12 |
| LD-DPO | 8.15 | 7.40 | 4.45 | 3.15 | 4.40 | 8.55 | 7.15 | 8.35 | 6.45 |
| **Llama3-8B-Instruct** | | | | | | | | | |
| SFT | 8.95 | 8.45 | 4.60 | 5.05 | 5.35 | 9.10 | 8.10 | 9.30 | 7.36 |
| DPO | 9.05 | 8.55 | 5.55 | 5.30 | 5.45 | 9.00 | 8.70 | 9.35 | 7.61 |
| R-DPO | 8.85 | 8.00 | 5.75 | 5.50 | 6.15 | 8.75 | 7.70 | 9.65 | 7.54 |
| SimPO | 9.05 | 7.40 | 5.45 | 5.30 | 5.75 | 8.60 | 7.90 | 9.45 | 7.36 |
| WPO | 8.20 | 8.55 | 5.50 | 5.20 | 6.20 | 9.25 | 8.50 | 9.40 | 7.60 |
| SamPO | 9.20 | 8.65 | 5.30 | 3.55 | 6.15 | 9.25 | 8.45 | 9.50 | 7.50 |
| LD-DPO | 8.95 | 8.35 | 5.90 | 5.35 | 6.75 | 8.70 | 8.55 | 9.40 | 7.74 |
| **Qwen2-7B-Base** | | | | | | | | | |
| SFT | 7.45 | 6.35 | 4.65 | 6.05 | 4.45 | 6.85 | 6.50 | 8.15 | 6.30 |
| DPO | 7.65 | 7.30 | 4.70 | 6.95 | 5.00 | 7.35 | 6.65 | 8.25 | 6.73 |
| R-DPO | 7.40 | 6.30 | 4.50 | 6.30 | 3.95 | 6.70 | 6.15 | 8.00 | 6.16 |
| SimPO | 7.75 | 6.60 | 4.70 | 5.95 | 5.45 | 6.60 | 7.60 | 8.30 | 6.61 |
| WPO | 7.85 | 6.75 | 4.85 | 6.75 | 5.10 | 6.75 | 7.25 | 8.45 | 6.71 |
| SamPO | 7.95 | 7.35 | 4.85 | 6.70 | 4.50 | 7.15 | 7.20 | 8.65 | 6.79 |
| LD-DPO | 8.20 | 6.85 | 5.05 | 7.45 | 4.85 | 6.70 | 6.80 | 8.50 | 6.80 |
| **Qwen2-7B-Instruct** | | | | | | | | | |
| SFT | 9.10 | 8.95 | 6.30 | 6.35 | 6.45 | 8.05 | 9.05 | 9.40 | 7.95 |
| DPO | 9.15 | 9.05 | 6.45 | 6.40 | 4.85 | 7.90 | 9.05 | 9.45 | 7.79 |
| R-DPO | 8.80 | 8.50 | 6.80 | 6.25 | 6.10 | 8.60 | 8.90 | 9.60 | 7.94 |
| SimPO | 8.85 | 8.90 | 6.20 | 6.55 | 6.00 | 8.00 | 9.05 | 9.50 | 7.88 |
| WPO | 9.00 | 8.80 | 6.80 | 6.40 | 4.90 | 8.10 | 8.20 | 9.55 | 7.72 |
| SamPO | 8.90 | 8.80 | 6.50 | 6.15 | 5.70 | 8.45 | 8.35 | 9.40 | 7.78 |
| LD-DPO | 8.90 | 8.55 | 7.25 | 6.25 | 5.90 | 8.75 | 9.05 | 9.55 | 8.03 |

Table 6: Scores for each capability item on the MT-Bench.

## B.3 MT-BENCH RESULT

We provide a detailed presentation of the MT-Bench results in Table 6. MT-Bench comprises 80 questions specifically designed to evaluate the model's proficiency across 8 dimensions: *writing, roleplay, reasoning, math, coding, extraction, stem, humanities*. To further assess the model's capability in multi-round interactions and bolster the validity of the results, each question in MT-Bench undergoes two rounds of Q&A. The scores reported for each dimension are the averages derived from these two rounds.

Generally, we find that LD-DPO outperforms SFT, DPO, and all other baselines on average in all settings. In the case of slight fluctuations in performance across other dimensions, LD-DPO significantly outperforms all baselines in reasoning, indicating that LD-DPO can enhance the reasoning capability of LLMs.

## C    IMPROVEMENT OF REASONING CAPABILITY

In further analysis of the MT-Bench, we found that the reasoning ability of the model significantly improved after applying LD-DPO, compared to both the SFT and DPO models. To further validate this conclusion, we conducted experiments on ProofWriter. ProofWriter is a specialized dataset designed to evaluate the reasoning capabilities of large language models. It comprises a broad range of problems, from direct reasoning tasks to those requiring more than five steps, and distinguishes between open-world assumptions (OWA) and closed-world assumptions (CWA), resulting in a total of 14 data subsets.

We conducted experiments on ProofWriter and the results are shown in Table.5. Fig.7 displays the results corresponding to the Llama3-8B-Instruct, with the scatter indicating the score for each data subset. After applying LD-DPO, the model shows an overall improvement across 14 data subsets. Compared to Llama3-8B-Instruct, the average score increases from 55.19 to **58.72**, outperforming five classes of preference optimization algorithms, including DPO, indicating a significant improvement in reasoning. Detailed results of MT-Bench shown in the Appendix B.3 also prove this point, and a case study of the reasoning problems in Appendix D.

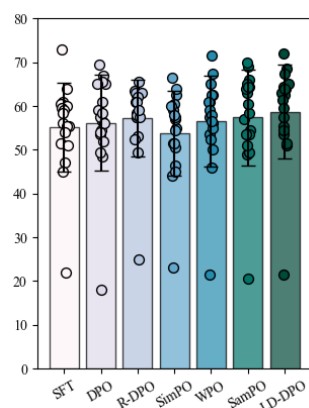

In fact, verbose responses negatively impact the reasoning abilities of language models, particularly smaller models. Unlike Chain of Thought (CoT), which is considered an excellent reasoning paradigm (Feng et al., 2024b), overly lengthy responses tend to include incorrect derivations or meaningless descriptions, which interfere with the model's ability to make the next step in the reasoning process or to reach the correct conclusion. Our approach enables the model to learn a concise CoT style while preventing overly lengthy answers, thereby improving the model's reasoning capabilities.

Figure 7: Performance of various methods on 14 subsets of ProofWriter dataset for the Llama3-8B-Instruct setting.

## D    CASE STUDIES

In Fig.8, we present two examples from AlpacaEval 2, where LD-DPO generates comparably accurate outputs with less tokens compared to vanilla DPO, showing that LD-DPO can generate more concise outputs without sacrificing performance. Since longer outputs are not necessarily more accurate or informative, sometimes they may contain more redundant and unimportant information, making the outputs verbose and lengthy. We believe that providing concise and clear outputs can also reflect the expressive capability of LLMs.

We also present two examples from ProofWriter in Fig.9, where LD-DPO generates more concise and direct chain-of-thought (CoT). From the cases we find that DPO generates complex and verbose CoTs, ultimately resulting in incorrect answers. By contrast, LD-DPO provides more direct and clearer CoTs, resulting in correct answers. The experimental results show that overly lengthy CoTs may detract from the accuracy of reasoning, whereas LD-DPO can improve the reasoning capability of LLMs by producing more concise and clearer CoTs.

| Input | Q: What language does Argentina people speak? |
|---|---|
| **Llama3-8B-Instruct DPO** (209 tokens) | In Argentina, the official language is Spanish. It's the most widely spoken language in the country, and it's used as the primary language for government, education, media, and daily life.

However, Argentina is a country with a rich cultural diversity, and you may also hear other languages being spoken, such as:
1. Italian: Many Argentines have Italian ancestry, and you may hear Italian being spoken, especially in the north of the country.
2. German: German is also a common language in Argentina, particularly in the Pampa region.
3. English: English is widely taught in schools, and many Argentines speak some English, especially in the service industry, tourism, and business.
4. Indigenous languages: There are also several indigenous languages spoken in Argentina, such as Quechua, Guarani, and Mapudungun, which are spoken by the country's indigenous communities.

Overall, Spanish is the dominant language in Argentina, and it's the language you'll most likely hear and use during your visit. |
| **Llama3-8B-Instruct LD-DPO** (72 tokens) | In Argentina, the official language is Spanish. It's the most widely spoken language in the country and is used as the primary language for government, education, media, and daily life. In fact, Argentine Spanish has its own unique dialect and accent, which is influenced by Italian, German, and other immigrant languages that have been incorporated into the culture over time. |

| Input | Q: Give a brief description of the given category of movies and shows: Period Dramas. |
|---|---|
| **Llama3-8B-Instruct DPO** (146 tokens) | Period dramas are a category of movies and shows that are set in a specific historical time period, often featuring elaborate costumes, settings, and storylines. These productions typically take place in the past, often in Europe or North America, and may be based on real events, literary works, or fictional stories. Period dramas often focus on the lives of aristocrats, royalty, or other members of high society, exploring themes such as social class, politics, romance, and personal struggles. The genre often features grand estates, opulent costumes, and meticulous attention to historical detail, transporting viewers to a bygone era. Examples of period dramas include "Downton Abbey", "Pride and Prejudice" and "The Crown". |
| **Llama3-8B-Instruct LD-DPO** (123 tokens) | Period Dramas refer to films and television shows that are set in a specific historical time period, often featuring elaborate costumes, settings, and cultural references from that era. These dramas typically recreate historical events, figures, or lifestyles with a focus on storytelling, character development, and atmospheric authenticity. They can range from epic historical epics to intimate character studies, and often explore themes such as love, power, class, and social change. Examples of Period Dramas include films like "Downton Abbey" and "Pride and Prejudice", as well as TV shows like "Outlander" and "Victoria". |

Figure 8: Comparing generations of AlpacaEval 2 prompts from Llama3-8B trained based on DPO and LD-DPO.

| | | |
|---|---|---|
| **Input** | Q: Anne is cold. Bob is big. Dave is rough. Gary is cold. Cold people are furry. Big people are rough. If someone is white then they are big. White people are cold. Red people are big. If someone is blue then they are white. Anne is furry? Options: A. True B. False | |
| **Llama3-8B-Instruct DPO** 
 **(231 tokens)** | 1. Anne is cold, so according to the given statement, cold people are furry. Therefore, Anne is furry. 
 2. Bob is big, but being big does not necessarily make someone furry. So, we cannot say whether Bob is furry or not. 
 3. Dave is rough, but being rough also does not indicate whether someone is furry or not. So, we cannot say whether Dave is furry or not. 
 4. Gary is cold, so he is furry according to the given statement. 
 5. The statement "Big people are rough" does not indicate whether big people are furry or not. So, we cannot say whether white people are furry or not. 
 6. The statement "White people are cold" does not indicate whether white people are furry or not. So, we cannot say whether red people are furry or not. 
 7. The statement "If someone is blue then they are white" does not indicate whether blue people are furry or not. 
 So, we cannot say whether Anne is furry or not. Therefore, we cannot conclude whether Anne is furry or not. The answer is "Cannot be determined." | |
| **Llama3-8B-Instruct LD-DPO** 
 **(43 tokens)** | Let's break it down step by step. 
 We know Anne is cold. From the statement "Cold people are furry", we can conclude that Anne is furry. 
 So, the answer is: A. True | |
| **Input** | Q: Bob is blue. Fiona is round. Harry is big. If someone is blue and cold then they are not big. Fiona is not big? Options: A. True B. False C. Unknown | |
| **Llama3-8B-Instruct DPO** 
 **(221 tokens)** | Let's break down the information given in the passage: 
 1. We are given the following statements: 
 * Bob is blue. 
 * Fiona is round. 
 * Harry is big. 
 2. We are also given a conditional statement: "If someone is blue and cold then they are not big." 
 3. We are asked to determine the truth value of the statement "Fiona is not big." 
 To do this, we need to analyze the given statements and the conditional statement. 
 From the given statements, we know that Fiona is round, but we don't have any information about her size. Therefore, we cannot conclude that Fiona is big or not big. 
 The conditional statement "If someone is blue and cold then they are not big" does not provide any information about Fiona's size or color. It only talks about someone who is both blue and cold, and in that case, they are not big. 
 Since we don't have any information about Fiona's size or color, we cannot conclude that she is not big. 
 Therefore, the correct answer is: B. False | |
| **Llama3-8B-Instruct LD-DPO** 
 **(139 tokens)** | Let's break it down step by step. 
 We are given the following statements: 
 1. Bob is blue. 
 2. Fiona is round. 
 3. Harry is big. 
 4. If someone is blue and cold then they are not big. 
 From statement 3, we know Harry is big. 
 The question asks if Fiona is not big. To determine this, let's analyze the information: We don't have any information about Fiona's size. The only characteristic we know about Fiona is that she is round (from statement 2). We don't have any information about her size being big or not big. 
 Therefore, the correct answer is: C. Unknown | |

Figure 9: Comparing generations of ProofWriter prompts from Llama3-8B trained based on DPO and LD-DPO.

