# OpenReview forum: "Length Desensitization in Direct Preference Optimization"
_ICLR.cc/2025/Conference — Submitted to ICLR 2025_

### Official Review · Reviewer_vvRF · 2024-10-29

**Soundness:** 2
**Presentation:** 4
**Contribution:** 3
**Rating:** 5
**Confidence:** 4

**Summary:**

This paper proposes LD-DPO, a DPO-based method to desensitize LLMs from length bias during preference optimization. The authors first gave analysis on why DPO methods are sensitive to the length bias in the response pairs. Based on the analysis, LD-DPO decays  the probability of the excessively long  part of responses to attenuate the sensitivity of DPO to longer responses. Results on several benchmarks demonstrate that compared to DPO, LD-DPO sucessfully reduces the length of generated responses after preference optimization.

**Strengths:**

1. Length bias widely exists in a wide range of LLM alignment methods and should be disentangled from real human preferences,

2. Motivation of LD-DPO is clearly expressed by the theoretical analysis.

3. Proposed method is evaluated on multiple benchmarks and base models.

**Weaknesses:**

1. Redundant symbol definitions. I do not think the definitions of $\mathcal{X}_1$, $\mathcal{X}_2$,$\mathcal{K}_1$,$\mathcal{K}_2$ are necessary. It just adds to the diffculty to understanding.

2. The colors in the fig. 3 are difficult to distinguish. And this figure is also a bit hard to comprehend.

3.Some spelling and grammartical mistakes, e.g. "Length **Desentsitization** of DPO, termed LD-DPO"

**Questions:**

1. The analysis of length bias in DPO is interesting. However, it seems this analysis only applies to DPO-based methods. Since RLHF-based methods also tend to increase generation length after training, how is it different from the length bias in DPO? Is it possible to apply your analysis to length bias in RLHF?

2. In LD-DPO, probabilities of excessively long portions (response after $l_p$) are decayed by $\alpha$ to close the gaps between the magnitudes of chosen and rejected responses' probabilities, which inevitbaly introduces information losses. And you also admitted that "additional text can convey more human-like preferences"; "$\alpha$ is actually the result of a compromise to achieve desensitization of DPO based on model capabilities and to prevent the loss of human-like preferences"

Therefore, the decrease of generation length is reasonable, but it is weird that LD-DPO consistently demonstrate better scores than DPO even with this information loss. Is there any reasonable explanation?

3. LD-DPO seems to be very sensitive to hyperparameter $\alpha$ (the values are different for all models in your experiments). Is there any way to improve it?

To be honest, I'm currently undecided between 5 and 6. Considering the issues mentioned above, I'll give this paper a 5 for now and reconsider it upon seeing authors' feedback.

---

> ### Author Response · Authors · 2024-11-19
> **Response to Reviewer vvRF (Part 1)**
>
> Thanks for your time in reviewing and providing feedback for our work! We are eager to further elaborate on our motivations and address your questions:
> ***
> **Weakness 1:** Redundant symbol definitions.
>
> **Response:** Thank you for pointing out our problem with the definition of symbols, which we will explain further：
> - In fact, by defining $\mathcal{X}_1, \mathcal{X}_2, \mathcal{K}_1, \mathcal{K}_2$, we hope to **enhance the simplicity of the subsequent formulas** and thus **improve the readers’ reading experience.** Otherwise, the formulas will look very cluttered as shown below:
> $$\mathcal{L}=-log(\frac{(\pi_r(y_l|x)\pi_t(y_w|x))^\beta}{(\pi_r(y_l|x)\pi_t(y_w|x))^\beta+(\pi_r(y_w|x)\pi_t(y_l|x))^\beta}).$$
> ***
> **Weakness 2:** The color scheme and meaning of Fig.3 is unclear.
>
> **Response:** Thank you very much for pointing out the problems we had with drawing the image. It was indeed our mistake, **we will redraw Fig.3 and the content description in the following link: http://gxwhy.net/ads/Fig3.pdf, and subsequently update them in the paper.**
>
> ***
> **Weakness 3:** Some spelling and grammatical mistakes.
>
> **Response:** Thank you for pointing out some of the spelling and grammatical issues in the article. **We will review the entire article to correct the original issues and apologize for any reading troubles.**
> ***

---

> ### Author Response · Authors · 2024-11-19
> **Response to Reviewer vvRF (Part 2)**
>
> **Question 1:** Generalization of analytical methods to the RLHF-base approach.
>
> **Response:** We are grateful for your valuable questions regarding our analytical methods! We will provide a more detailed elaboration based on your questions.
> - As you mentioned, our analysis is applicable to most DPO-based methods, like SimPO, R-DPO, etc. This is because these methods need $y_w$ and $y_l$ as preference data pair, and **fitting implicit rewards based on model-predicted probabilities** can result in length sensitivity.
> - For RLHF-based methods, we review the unified optimization paradigm for RLHF, where $\pi_a$ is actor model and $\pi_r$ is reference model:
> $$J_r(\pi_a)=\mathbb{E}_D[r(x,y)-\beta\log\frac{\pi_a(y|x)}{\pi_r(y|x)}]$$
>    Intuitively, the gradient of the RLHF-based methods depends on the reward $r(x,y)$ and the KL divergence term, compared with the former, the latter has a negligible correlation with the data length.
> - As described in numerous papers[1-3], the length bias issue of the RLHF-based approach arises from the length preference of the **reward model itself**, which is different from the DPO-based methods described above. Regarding this problem, we believe that a correction to the training process can be achieved by adding a length-related penalty term to the reward part. (There is already relevant work concerning this approach [3-4])
>
> Overall, our method is indeed more suitable for achieving length desensitization in DPO-based methods. For RLHF-based methods, penalizing the length of the reward component might be simpler and more effective.
> ***
> **Question 2:** Comparison of LD-DPO and DPO performance.
>
> **Response:** We are grateful for your careful reading of the experimental analysis section in the article as well as your meaningful questions! We will explain the performance issues of LD-DPO in detail.
> - In Section 3.1 of the article, we conduct a detailed analysis of the length sensitivity of the DPO. It seems that DPO has no information loss. However, the severe length sensitivity of the optimization objective causes its mathematical modeling to fail to effectively represent the real information.
> - By remodelling the likelihood, LD - DPO alleviates the impact of length. It seemingly discards certain information, thus enabling the optimization process to proceed in a more accurate direction.
> - It is an undeniable fact that the performance of LD-DPO deteriorates when the value of  $\alpha$ is chosen close to 0. This is because, in such a case, it actually discards some of the information within the data.
>
> Overall, LD-DPO can be regarded as a trade-off between an **"inaccurate optimization direction(DPO)"** and the **"loss of some information($\alpha=0$)"**, neither of which is advantageous. This explains the inverted U-shape of the performance curves shown in Fig.4. Consequently, LD-DPO surpasses DPO on numerous benchmarks. Moreover, you can also refer to the case study in Appendix D.
>
> ***
> **Question 3:** Hyperparameter sensitivity of LD-DPO.
>
> **Response:** Thank you very much for your valuable question and we will explain further about the hyperparameter sensitivity of the LD-DPO.
> - Undeniably, the performance of LD-DPO is indeed influenced by the hyperparameter $\alpha$, presenting an overall inverted - U shape across many benchmarks..
> - To be honest, our experiments revealed that for the same model, making an arbitrart selection of $\alpha$ across a wide span(for instance, in the case of Llama3-8B-Instruct, within in range of [0.3, 0.7]) almost invariably leads to better performance than that of DPO. This indicates that LD-DPO has good robustness in performance with respect to the $\alpha$
> - Our analysis of the various capability models in Section 5.2 shows that the length sensitivity of DPO is relevant to the model's capabilities. Researchers can select the appropriate $\alpha$ based on the model’s capabilities. For example, $\alpha$ near 0.6 can be chosen for a recent 7B LLM, and $\alpha$ near 0.8 can be chosen for a recent 70B LLM.
>
> Overall, LD-DPO is indeed sensitive to $\alpha$. However, it also provides a wider range of options at the same time. **Moveover, we will provide the recommended intervals for the different capability models currently available in a subsequent version of the paper.**
>
> ***
> **Thank you again for your valuable comments on our work, and we hope that our exlanation could address your concern!**
> ***
>
>
> [1] Disentangling length from quality in direct preference optimization
>
> [2] Rethinking LLM-based Preference Evaluation
>
> [3] Post-hoc reward calibration: A case study on length bias
>
> [4] Improving alignment of dialogue agents via targeted human judgements

---

> ### Author Response · Authors · 2024-11-22
> **Follow-Up on Rebuttals**
>
> We want to sincerely thank the reviewers for their time and effort in evaluating our paper. We would appreciate it if you could kindly confirm that the rebuttal was received and let us know if any additional steps or clarifications are required from our side. Your feedback is highly important to us, and we remain available to address any further concerns or questions.
>
> Please let us know. Thanks.

---

> > ### Comment · Reviewer_vvRF · 2024-11-24
> >
> > Thanks for the authors' responses. After reading the responses and the discussions with other reviewers, I have decided to keep my score due to issues like sensitivity to hyperparameteres and novelty.

---

### Official Review · Reviewer_Jjwf · 2024-10-29

**Soundness:** 3
**Presentation:** 4
**Contribution:** 2
**Rating:** 5
**Confidence:** 4

**Summary:**

This paper addresses the issue that DPO tends to over-optimize for verbosity and proposes a method to desensitize DPO to data length. Evaluations show the proposed LD-DPO algorithm consistently outperforms existing algorithms with less average tokens than DPO.

**Strengths:**

1. Addresses a popular issue of DPO's sensitivity to length.
2. Good presentation and easy to read.
3. Good empirical performance.

**Weaknesses:**

1. Although theoretical insights on why DPO favors longer response is provided, the proposed LD-DPO is a heuristic method. It directly cuts off the importance of the tokens exceeding the public length. It is disappointing to see the solution to the well-formulated length sensitivity problem is just a code-level heuristic method. Why not try to modify the DPO loss for a loss landscape[1] that is length-desensitized?
2. The description for eq.(10) is not rigorous. Why $p^\alpha$ is "human-like preferences" and  $p^{1-\alpha}$ is "verbosity preference"? Just by definition?

Overall, it is just another LLM paper following DPO. There's nothing particularly exciting, and there isn't much to comment on. While it does not present anything particularly novel or insightful, it is a well-structured paper with a thorough evaluation.

[1] Towards Analyzing and Understanding the Limitations of DPO: A Theoretical Perspective

**Questions:**

See weekness.

**Details Of Ethics Concerns:**

There are no apparent violations of the code; however, it is worth noting that this paper is available on Arxiv.

---

> ### Author Response · Authors · 2024-11-20
> **Response to Reviewer  Jjwf**
>
> Thank you for taking the time to review and provide feedback on our work! We are glad to address your questions and provide further clarification on our research:
> ***
> **Weakness 1:** LD-DPO is just a code-level heuristic method.
>
> Thank you very much for recognizing our analysis of the DPO length sensitivity issue and the questions raised about the design of the LD-DPO method, which we will explain below.
>
> - Our theoretical analysis indicates that the length sensitivity of DPO stems from the issue of the $\log\sigma(r_w-r_l)$ function, which is utilized in DPO as a means of fitting preference distribution. Theoretically, we could significantly modify the fitting function to avert the length sensitivity. In fact, there are many methods available using other functions such as RRHF, KTO, IPO. However, their overall performance is less satisfactory compared to that of DPO (refer to [1]). Hence, we assert that the $\log\sigma(r_w-r_l)$ function is still the optimal fitting function at present.
> - Early in our experimental process, we contemplated a formal fine-tuning of the $\log\sigma(r_w-r_l)$ function with the aim of alleviating the length sensitivity of the DPO, but the outcomes did not prove to be satisfactory. In fact, both R-DPO[2] and SimPO[1] hope to alleviate this problem by fine-tuning the form. However, through a theoretical derivation similar to Eq.5-Eq.7, it can be observed that incorporating a constant term related to the length of the preferred data pair has no impact on the relationship between the gradients. Since the loss function will continue to assume the form of the following formula, the conclusion drawn from Equation 7 still remains valid. Therefore, we have abandoned this idea.
>   $$ \mathcal{L}=-log\sigma(\beta\log\frac{\pi(y_w|x)}{K_1}-\beta\log\frac{\pi(y_l|x)}{K_2}) $$
> - In fact, LD-DPO is a concise and efficient approach, which addresses the root cause underlying DPO's length sensitivity, corrects the flaws present in the original DPO algorithm, and attains good outcomes.
> ***
> **Weakness 2:** The description for Eq.10 is not rigorous.
>
> Thank you very much for your valuable question and we will explain Eq.10 further.
>
> - The aim of preference optimization is to enable the model to learn human preferences within the data. Whether the preferences are labeled manually or by an LLM, length preference is an aspect of these preferences. What this paper intends to mitigate is the significant sensitivity issue of DPOs with respect to length preferences. Consequently, it is desirable to separate the length factor from the modeling process.
> - After decoupling the length preference, the likelihood can then be naturally split into two parts, namely the length preference part and the other part (which, within the context of this paper, represents the human-like preferences part). Viewed from another angle, both the variation in token content and the quantity of tokens will exert an influence on the likelihood. Consequently, Eq.10 serves as an intuitive mathematical model for us.
> ***
> Overall, LD-DPO can be seen as an enhancement of the DPO algorithm. Currently, the DPO algorithm has gained extensive acceptance among researchers and is widely utilized in the alignment phase of Large Language Models (LLMs). Nevertheless, numerous researchers[2-4] have noticed that LLMs after undergoing DPO processing tend to have the issue of output redundancy, and several solutions have been put forward[1-2]. However, the current research on this particular problem remains unclear, which has led to these proposed methods failing to achieve satisfactory outcomes.
>
> Our work alleviates this problem through conducting a profound theoretical analysis of DPO. We contend that DPO is afflicted with a length sensitivity issue and, based on this, we have devised a concise and efficient LD-DPO algorithm. As shown in the paper, LD-DPO demonstrates outstanding results across multiple benchmarks. It mitigates the problem of redundant replies in post-DPO LLMs and attains a better alignment with human preferences.
> ***
> **Thank you again for taking the time to review and provide valuable feedback on our work, and we hope that our exlanation could address your concern!**
>
> [1] SimPO: Simple Preference Optimization with a Reference-Free Reward
>
> [2] Disentangling length from quality in direct preference optimization
>
> [3] Rethinking LLM-based Preference Evaluation
>
> [4] Post-hoc reward calibration: A case study on length bias

---

> ### Comment · Reviewer_Jjwf · 2024-11-20
>
> Thanks for the response. I still have the following concerns:
>
> # About response to weakness 2
>
> This still does not answer why the decoupling defined in Eq.10 can effectively decouple the "human-like" part and the "verbosity" part. Why could it not be an addition instead of multiplication? My point is that the formulation and "human-like""verbosity" descriptions are not rigorous, especially when you need prior knowledge for $\alpha$ to determine how much of "human-like" part preference there is.
>
> # About response to weakness 1
>
> It seems that your point is that the methods with better theoretical guarantees often result in unsatisfactory empirical performance. That's why you propose this simple and better-performed approach.
>
> I have an idea, to desensitize length, why not just average the likelihood of response by the length like the formulation below?
> $$\hat{\pi}=\pi^{-l}(y|x)$$
> Just taking the power of negative length $-l$ could result in a strictly length-desensitized likelihood, which is much more concise and intuitive than your proposed formulation (Eq.11).  And unlike your formulation, this formulation will not result in information loss for the tokens exceeding the public length.

---

> > ### Author Response · Authors · 2024-11-21
> > **Response to Reviewer Jjwf**
> >
> > Thank you very much for your response and we will explain in response to your concerns.
> > ***
> > **Weakness 2:** The description for Eq.10 is not rigorous.
> >
> > **Response:** Thank you very much for your valuable question and we will explain Eq.10 further.
> >
> > 1. First, it is recognized that $\pi(y|x)$ is related to the content of the tokens, the number of tokens. As an example, when $y_w$ and $y_l$ have the same length, the difference between $\pi(y_w|x)$ and $\pi(y_l|x)$ comes from the content of the tokens completely, and conversely, receives the effect of the length.
> > 2. Our goal is to reduce the impact of the latter, so we propose this decoupling method. As to why we choose multiplication instead of addition because:
> >    1. The calculation of probability is itself a cumulative multiplication of probabilities, it is quite logical to use multiplication.
> >    2. By means of decoupling via the use of multiplication along with power arithmetic degradation, performing the log function operation on the formula will cause it to become more concise and elegant.
> > 3. We really can not accurately determine in advance how much of the impact of the two parts, which is why we need a hyperparameter $\alpha$, but **as long as a certain range of decoupled part of the length preference, is able to optimize the direction of the positive gains (experiments have also proved that a large range of values of our methods are better than DPO)**
> > 4. With your correction, we realize that the term “human-like” may not be entirely appropriate, as we are unable to determine at this time whether there are any undesirable elements other than length. We will change “human-like preference” to “other preference” in a subsequent version to make it clear that this paper is only about length preference. Thanks again for your correction!
> > ***
> > **Weakness 1:** Length decoupling method of LD-DPO.
> >
> > **Response:** **"The methods with better theoretical guarantees often result in unsatisfactory empirical performance." is not our opinion. Our opinion is that "LD-DPO is a concise and efficient method for alleviating the length sensitivity issue of DPO".** In fact, we have tried a number of preference optimization methods, which will be explained in more detail below in response to your question.
> >
> > - We are not arguing that methods with better theoretical guarantees do not perform well enough, and in fact they all drive the development of offline preference optimization.
> >   - In our previous response, RRHF, KTO, and IPO can be regarded as contemporaneous work of DPO. They proposed different methods of modeling preferences compared to DPO and provided theoretical guarantees to contribute to offline preference optimization.
> > - Subsequent researchers have identified the shortcoming of DPO, i.e., the phenomenon of redundant responses, and proposed some intuitive solutions, such as SimPO and R-DPO. Although they modified the loss function of DPO, the length desensitization of these methods is unsatisfactory because they do not analyze the phenomenon theoretically.
> > - LD-DPO is a concise and effective method that we designed after theoretically proving the length sensitivity of DPO. It is more to the point than the former.
> > - The idea you mentioned is indeed a concise and intuitive method, and there is indeed such a paper recently[1].
> >   - In fact, compared to DPO, this approach completely alters the likelihood modeling and differs more from the original $\pi(y|x)$, **resulting in a greater loss of information rather than “no loss of information”.**
> >   - You can refer to Figure1-2 in [1], where you can see that the change in the response length for this method is not significant. In addition, we read this paper subsequently and carried out related experiments. After choosing its recommended parameters, **the performance of Llama3-8B-Instruct on AlpacaEval 2 is shown in the following table . We will add this method as a baseline in the next version of the paper.**
> >
> >     | Method | LC-Winrate(%) | Avg.Token |
> >     |---|---|---|
> >     | DPO | 40.21 | 393 |
> >     | LN-DPO[1] | 40.56 | 366 |
> >     | **LD-DPO** | **44.00** | **308** |
> >     | | |
> >
> > Thank you very much for seriously discussing the length decoupling method with us, and we hope that our response will address your concern.
> > ***
> > **Thank you again for taking the time to review and provide valuable feedback on our work, and we hope that our exlanation could address your concern!**
> >
> > [1] The Hitchhiker’s Guide to Human Alignment with *PO

---

> > > ### Comment · Reviewer_Jjwf · 2024-11-21
> > >
> > > Thanks for the reply.
> > >
> > > It seems that the method in [1] is not what I meant, they are dividing the likelihood by length but I meant to take the power of negative length. But thank you for the detailed explanation.
> > >
> > > Empirical performance can be the result of many different reasons rather as information loss, it could be due to the magnitude of modified likelihood, hyperparameters, or even code-level factors. In your formulation, however, you explicitly reduce the importance of later tokens. While this may work for the current datasets, it is inherently prone to information loss for practical scenarios that require consideration of later tokens.
> > >
> > > Additionally, the theories regarding the length sensitivity of DPO heavily rely on prior results [2], and you even use the same notations. As such, your theoretical contribution is incremental and does not sufficiently strengthen the overall contribution.
> > >
> > > ***
> > > I think I have gathered enough information and will now wait for the discussion with the other reviewers to conclude.
> > >
> > > *[1] The Hitchhiker’s Guide to Human Alignment with PO*
> > >
> > > *[2] Towards Analyzing and Understanding the Limitations of DPO: A Theoretical Perspective*

---

> ### Author Response · Authors · 2024-11-21
> **Response to Reviewer Jjwf**
>
> Thank you very much for your response, we consider that a further explanation is necessary.
>
> - In fact, the method in [1] takes the power of $\frac{1}{l}$, just using a different form like this:
>   $$\log \pi^{\frac{1}{l}}(y|x) = \frac{1}{l}\log \pi(y|x)$$
> - Our scenario is to improve the comprehensive conversational ability of LLMs, and the preference optimization dataset we chose is UltraFeedback[2], which is designed to align with human preferences and improve the helpfulness of the model. This dataset is representative of the alignment phase and matches the actual scenario, which is enough to prove the effectiveness of the method.
> - We cite the theoretical approach in [3] with partial corrections. However, our contribution is to give a complete proof of the DPO length sensitivity, to explain theoretically this problem recognized by researchers, and to propose a efficient method LD-DPO. These are not mentioned in [3].
> ***
> Thank you very much for taking the time to engage in a discussion with us, and we hope that our response will give you further insight into the motivations and contributions for our work.
>
> [1] The Hitchhiker’s Guide to Human Alignment with *PO
>
> [2] UltraFeedback: Boosting Language Models with Scaled AI Feedback
>
> [3] Towards Analyzing and Understanding the Limitations of DPO: A Theoretical Perspective

---

> > ### Comment · Reviewer_Jjwf · 2024-11-21
> >
> > About the formulation
> >
> > "$\log \pi^{-l}$ should be $-l \log \pi$", not "$-\frac{1}{l} \log \pi$"
> >
> > Additionally, the formulation in eq.(1) of [1] is neither one of the two expressions mentioned above, instead, it is "$ \log \frac{\pi}{l}$"
> >
> > Thanks for the response.

---

> ### Author Response · Authors · 2024-11-21
> **Response to Reviewer Jjwf**
>
> Thank you for your reply, we did have a problem with the last reply.
>
> We reread the paper [1], and its methodology is actually as in Table 2 like this:
> $$\log \pi^{\frac{1}{l}}(y|x) = \frac{1}{l}\log \pi(y|x)$$
> As for your pointing out that Eq.1 is actually the author comparing it to the $\gamma$ in SimPO[2].
>
> This does differ from your idea, but there may be some similarities. Since both $-l$ and $\frac{1}{l}$ are exponential weightings that tend to have a trend opposite to the length of the data. However, $-l$ may lead to a change in the sign of the optimization objective, and we will conduct experiments related to your idea to verify its validity.
>
> We truly appreciate you taking the time to discuss our work with us and provide many valuable comments！
>
> [1] The Hitchhiker’s Guide to Human Alignment with *PO
>
> [2] SimPO: Simple Preference Optimization with a Reference-Free Reward

---

### Official Review · Reviewer_Rxjk · 2024-10-31

**Soundness:** 2
**Presentation:** 2
**Contribution:** 2
**Rating:** 5
**Confidence:** 4

**Summary:**

The authors demonstrate the existence of length sensitivity in the DPO algorithm and analyze this issue theoretically. They propose the LD-DPO algorithm to address this sensitivity. Experiments on three open-source language models show the effectiveness of LD-DPO.

**Strengths:**

- The logic flow is clear.
- The authors identify the reason for length sensitivity in the DPO algorithm.
- Based on their analysis, the authors propose the LD-DPO algorithm, which performs well in terms of length control and alignment.
- Experiments with three models across two datasets demonstrate the generalizability of LD-DPO.
- LD-DPO is a simple yet effective method.

**Weaknesses:**

- Although the authors claim to have theoretically proven the sensitivity of DPO to length, the description is still insufficiently rigorous. For example, from Equation 4 to Equation 5, the expectation sign is omitted without further explanation.
- The explanation from lines 211 to 215 is vague and overly intuitive, especially regarding the relationship between length and probability.
- In Equation 7, the authors take the absolute value of the ratio of two Jacobians, a less clear motivation that complicates the analysis.
- The term "probability bias" in line 416 is unclear.
- The effect of the hyperparameter \alpha remains unclear. In lines 497-500, the authors state, "Conversely, when \alpha is too small, excessive length decoupling leads to a loss of human-like preferences in the text, thereby reducing the optimization effectiveness." Figure 4 shows that different selections for \alpha lead to varied effects across different experiments. In AlpacaEval 2, choosing either 1 or 0 results in similar LC-win rates; however, in MT-Bench, choosing 0 (i.e., strong desensitization to length) leads to significantly lower performance compared to the original DPO. The authors do not provide further explanation for this.

- The design of the hyperparameter \alpha has a relatively strong impact on LD-DPO performance, as reflected by the results in Figure 4.

- A minor issue: the color differentiation of the lines within the same subfigure in Figure 3 makes it difficult for readers to distinguish them.

**Questions:**

Are lines 808-810 correct? Should both $\frac{\partial \mathcal{L}{DPO}(\chi_1; \chi_2)}{\partial \chi_1}$ and $\frac{\partial \mathcal{L}{DPO}(\chi_1; \chi_2)}{\partial \chi_2}$ increase when $\chi_2$ decreases? Should $\frac{\partial \mathcal{L}{DPO}(\chi_1; \chi_2)}{\partial \chi_1}$ decrease and $\frac{\partial \mathcal{L}{DPO}(\chi_1; \chi_2)}{\partial \chi_2}$ increase when $\chi_1$ decreases?

---

> ### Author Response · Authors · 2024-11-20
> **Response to Reviewer Rxjk (Part 1)**
>
> Thank you for taking the time to review and provide feedback on our work! We are glad to address your questions and provide further clarification on our research:
> ***
> **Weakness 1:** Eq.4 to Eq.5 is not rigorous.
>
> **Response:** Thank you very much for pointing out our problems with formula writing, we will explain further:
>
> - We apologize for lacking a more detailed explanation of the process from Eq.4 to Eq.5. We will add an explanation similar to “Assuming that only $y_w$ and $y_l$ are used to approximate the above expectation in the case of identically distributed data, gives us the following empirical estimation, ...” in the next version of the article to clarify why the expectation sign was omitted. Indeed, numerous related papers, such as [1-3], adopt this approach as it contributes to a more concise presentation of the paper.
> ***
> **Weakness 2:** Lines 211 to 215 is vague and overly intuitive.
>
> **Response:** Thank you very much for your valuable suggestions on our rationale section. We will re-explain it.
>
> 1. Through the analysis within the paper, we have reached the conclusion in Eq.7 that the absolute magnitude of the gradient in the two optimization directions of the DPO depend on the predicted probabilities of $y_w$ and $y_l$ by the actor model.
> 2. The DPO algorithm uses sentence-level predicted probability, calculated as:
>    $$\pi(y|x)=\prod_{i=1}^{len(y)}p(y_i|x,y_{<i})$$
>    where $y_i$ is the i-th token in $y$ and $y \in \{y_w, y_l\}$. Furthermore, since $p(y_i|x,y_{<i})\in[0,1]$, the large probability that $\pi(y|x)$ is smaller when $y$ is longer, which is indeed intuitive, and we give a sideways proof of this in Figure 2.
> 3. Combining the analysis in 2 with Figure 2, we can conclude that in the general case:
>    1. When $y_w$ is longer than $y_l$, the DPO optimization objective has a larger gradient in the $y_w$ direction.
>    2. When $y_l$ is longer than $y_w$, the DPO optimization objective has a larger gradient in the $y_l$ direction.
> 4. In lines 214 to 215, we hope to explain why this problem causes lengthy output from the post-DPO model:
>    1. When DPO goes to increase the probability of a longer $y_w$, which is a directed optimization, the length increase is obvious.
>    2. When DPO goes to decrease the probability of a longer $y_l$, which is not a directed optimization. We don't know if the output will be shorter, but it is a fact that there is a missed opportunity to directionally optimize for shorter $y_w$.
>
> **The above content is what the text intends to convey in lines 211 to 215, with an emphasis on the causes of length sensitivity of DPO and its resulting impact. We hope that our explanation can address your concern!**
> ***
> **Weakness 3:** Motivation of Eq.7 is unclear.
>
> **Response:** Thank you for your valuable question and we will explain the motivation for Eq.7.
>
> Obviously, to reduce the loss of DPO, the model can either choose to increase $\pi(y_w|x)$ or decrease $\pi(y_l|x)$. Therefore, the ratio of the gradient values of these two can reflect the tendency of the model in these two optimization directions as well as the influencing factors. It is worth noting that for this analytical process, we refer to [3], which is a paper analyzing DPO theory, although it does not address length sensitivity.
> ***
> **Weakness 4:** "Probability bias" in line 416 is unclear.
>
> **Response:** Thank you very much for pointing out our clerical error in line 416. We intended to use “probability difference” instead of what was written there. **We apologize for any ambiguity this may have caused.**
> ***
> **Weakness 5:** The effect of the hyperparameter $\alpha$ remains unclear.
>
> **Response:** Thank you for your valuable questions and we will explain further about Figure 4.
>
> In fact, the reason for the different performance of LD-DPO on MT-Bench and AlpacaEval 2 when $\alpha=0$ comes from the **different evaluation metrics**.
>  - For MT-Bench, the evaluation metric is **"score"**, i.e., the responses are evaluated using the judge model(such as GPT4). In this situation, when excessive information is lost during training and the output is overly short, the score is lower than that of DPO. Moreover, the judge model itself has a preference for long responses, which is one of the reasons for the significant drop in the score.
>  - For AlpacaEval 2, the evaluation metric is **"length-controlled winrate"**. The existence of a length-related penalty factor within the computational equation will, to a certain extent, counterbalance the length preference of the judge model. Consequently, when $\alpha=0$(i.e., when the answer length is at its shortest), LD-DPO will also exhibit good performance on AlpacaEval 2. You can refer to [4] for more detailed information about AlpacaEval 2.

---

> ### Author Response · Authors · 2024-11-20
> **Response to Reviewer Rxjk (Part 2)**
>
> **Weakness 6:** Hyperparameter sensitivity of LD-DPO.
>
> **Response:** Thank you very much for your valuable question and we will explain further about the hyperparameter sensitivity of the LD-DPO.
> - Undeniably, the performance of LD-DPO is indeed influenced by the hyperparameter $\alpha$, presenting an overall inverted - U shape across many benchmarks..
> - To be honest, our experiments revealed that for the same model, making an arbitrart selection of $\alpha$ across a wide span(for instance, in the case of Llama3-8B-Instruct, within in range of [0.3, 0.7]) almost invariably leads to better performance than that of DPO. This indicates that LD-DPO has good robustness in performance with respect to the $\alpha$
> - Our analysis of the various capability models in Section 5.2 shows that the length sensitivity of DPO is relevant to the model's capabilities. Researchers can select the appropriate $\alpha$ based on the model’s capabilities. For example, $\alpha$ near 0.6 can be chosen for a recent 7B LLM, and $\alpha$ near 0.8 can be chosen for a recent 70B LLM.
>
> Overall, LD-DPO is indeed sensitive to $\alpha$. However, it also provides a wider range of options at the same time. **Moveover, we will provide the recommended intervals for the different capability models currently available in a subsequent version of the paper.**
> ***
> **Weakness 7:** The color scheme of Figure 3 is inappropriate.
>
> **Response:** Thank you very much for pointing out the problems we had with drawing the image. It was indeed our mistake, **we will redraw Figure 3 and the content description in the following link: http://gxwhy.net/ads/Fig3.pdf, and subsequently update them in the paper.**
> ***
> **Question 1:** Derivation problem for lines 808 to 810.
>
> **Response:** Thank you very much for carefully reading our Appendix section and asking valuable questions! We will provide further explanations.
>
> In fact, we are solving for the partial derivatives in Eq.18-Eq.21 for the actual values, while lines 808-810 are analyzed for the absolute value of the gradient. **Since the result in Eq.15 is negative, the increasing or decreasing trend may change when the absolute value is taken.** You can re-check our content and please keep correcting us if you still have doubts.
> ***
> **Thank you again for your valuable comments on our work, and we hope that our exlanation could address your concern!**
> ***
>
> [1] Towards a unified view of preference learning for large language models: A survey
>
> [2] SimPO: Simple Preference Optimization with a Reference-Free Reward
>
> [3] Towards analyzing and understanding the limitations of dpo: A theoretical perspective
>
> [4] Length-Controlled AlpacaEval: A Simple Way to Debias Automatic Evaluators

---

> ### Author Response · Authors · 2024-11-22
> **Follow-Up on Rebuttals**
>
> We want to sincerely thank the reviewers for their time and effort in evaluating our paper. We would appreciate it if you could kindly confirm that the rebuttal was received and let us know if any additional steps or clarifications are required from our side. Your feedback is highly important to us, and we remain available to address any further concerns or questions.
>
> Please let us know. Thanks.

---

> ### Comment · Reviewer_Rxjk · 2024-11-22
>
> Thanks for the response,
>
> I've read them all. I encourage authors to revise their manuscripts directly.
> For me, the most interesting part of this submission is the analysis of equations 5-7.  Reviewer Jjwf pointed out that these analyses heavily rely on the paper [1]. This raises concerns about the novelty of the submission. I decided to keep the current rating.
>
> [1] Towards Analyzing and Understanding the Limitations of DPO: A Theoretical Perspective

---

> > ### Author Response · Authors · 2024-11-23
> > **Response to Reviewer Rxjk**
> >
> > Thank you for your response, we would like to further indicate our contibution to address your concerns.
> >
> > - One of our contributions is to **theoretically prove the length sensitivity of DPO and analyze the impact of length preferences on the direction of DPO optimization**. This is **totally different** from the motivation and contribution of [1], **which in fact does not contain any word related to “length” at all, and does not have any similar analysis and experiments.** Therefore, our contribution at the theoretical level cannot be denied because of [1].
> > - The redundancy of model output after DPO is a problem widely recognized by researchers, but **there has been no relevant theoretical proof until our work**, so it leads to the fact that there has not been an effective method. Our work defines this problem theoretically and proposes an efficient method, LD-DPO, **whose efficiency is verified through extensive experiments.**
> > ***
> > We sincerely appreciate the time and effort you have taken to review our work. We hope that with a closer re-examination of the essential differences between our work and [1], the motivations and contributions of our study can become clearer. Your feedback is invaluable, and we genuinely look forward to further discussions to better clarify our work.
> >
> > [1] Towards Analyzing and Understanding the Limitations of DPO: A Theoretical Perspective

---

### Meta-Review · Area_Chair_JVhs · 2024-12-22

**Metareview:**

This paper studies the verbosity bias of the DPO algorithm. It investigates the reasons for verbosity and provides solutions with empirical evaluation.

Strengths:
This paper is well-written and easy to understand. The proposed algorithm, LD-DPO, works well on selected tasks.

Weaknesses:
The biggest weakness of this paper seems to be the generality of the results. The analysis of verbosity bias is limited to the DPO algorithm, and the proposed LD-DPO is heuristic. Also, some reviewers are concerned about the novelty of the methodology in this paper.

Over the review and rebuttal period, none of the reviewers were excited about this paper. I agree with their comments and vote to reject.

**Additional Comments On Reviewer Discussion:**

During the rebuttal period, the discussion mainly focused on the novelty and generality of the results; however, these concerns were not addressed in the end.

---

### Decision · Program_Chairs · 2025-01-22

Reject